# VARIATIONAL INFERENCE FOR CYCLIC LEARNING

**Zhuojun Zou[1], Jie Hao[1,2]**
[1]Institute of Automation, Chinese Academy of Sciences, Beijing, China
[2]Guangdong Institute of Artificial Intelligence and Advanced Computing, Guangzhou, China
`{zouzhuojun2018, jie.hao}@ia.ac.cn`

## ABSTRACT

Cyclic learning has emerged as a powerful paradigm for weakly-supervised learning. It involves training with pairs of inverse tasks and leverages cycle-consistency in the design of loss functions. However, its potential remains underexplored, as current methods are often narrowly focused on domain-specific implementations. In this work, we develop generalized solutions for both pairwise cycle-consistent tasks and self-cycle-consistent tasks. By formulating cross-domain mappings as conditional probability functions, we reformulate the cycle-consistency objective as an evidence lower bound optimization problem via variational inference. Based on this formulation, we further propose two training strategies for arbitrary cyclic learning tasks: single-step optimization and alternating optimization. Our framework demonstrates broad applicability across diverse tasks. In unpaired image translation, it offers a theoretical justification for CycleGAN and yields CycleGN—a competitive GAN-free alternative. In unsupervised tracking, following our conceptual design, CycleTrack and CycleTrack-EM achieve state-of-the-art results on multiple benchmarks. This work establishes the theoretical foundations of cyclic learning and offers a general paradigm for future research. The source codes for CycleGN and CycleTrack are publicly available.

## 1 INTRODUCTION

The need for labeled data is now one of the biggest obstacles in machine learning research, where supervised learning's reliance on manual labeling introduces both scalability issues and quality control challenges. To address this, researchers have turned to self-supervised training, the core idea of which is to generate supervisory signals from unlabeled data for training. **Self-consistency**-based self-supervised learning has already demonstrated strong capabilities in the field of representation learning (Zhang et al., 2016; He et al., 2022; Mikolov et al., 2013; Devlin et al., 2019; Chen & He, 2021). A series of studies have now shifted focus to cross-domain self-supervised learning constructed via **cyclic consistency** (Xu et al., 2023; Yuan et al., 2020; Dwibedi et al., 2019; Wang et al., 2024; Kulkarni et al., 2019).

This type of approach involves designing a pair of inverse tasks and constructing the training process by leveraging the property that data points should return to their origin after cyclic processing. This not only eliminates the reliance on manual annotations but also preserves task-specific semantic constraints. As shown in Fig. 1, this framework has been applied to various tasks (Zhu et al., 2017; Wang et al., 2024; 2019b; Dwibedi et al., 2019). A well-known example is CycleGAN (Zhu et al., 2017) (Fig. 1(a)), which jointly optimizes two tasks by combining cycle consistency loss and adversarial loss, leading to its widespread adoption in various weakly supervised visual tasks (Almahairi et al., 2018; Yang et al., 2020; Kwon & Park, 2019). In contrast, a different approach is employed in the visual grounding (Referring Expression Comprehension) and image caption (Referring Expression Generation) loop (Fig. 1(b)), where both CyCO (Wang et al., 2024) and SC-Tune (Yue et al., 2024) adopt alternating training strategies for the two tasks, showcasing the cross-modal adaptation capability. In CyCO, they first conduct cyclic training using only the cross-entropy loss with image captioning as the objective, followed by another training batch that utilizes only the bounding-box losses (Rezatofighi et al., 2019; Girshick, 2015) for visual grounding. However, current approaches face two key limitations: First, task-specific designs hinder cross-domain generalization (e.g., the loss of CycleGAN cannot be directly applied to the video alignment task). Second, many methods

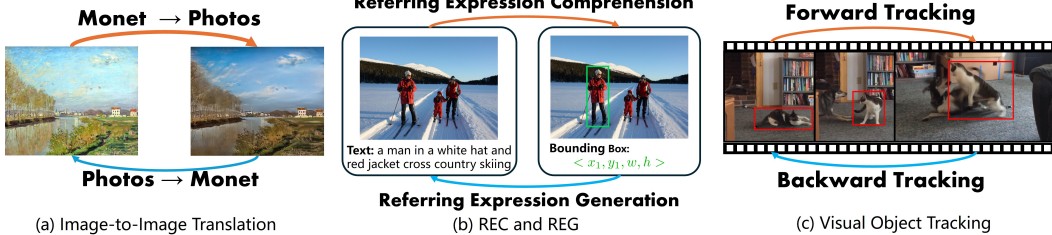

Figure 1: Tasks forming cyclic learning. (a) Image-to-image translation: forms cycles through paired style transformations (e.g., photo↔painting). (b) Referring expression comprehension & referring expression generation: construct cycles by bridging positional descriptions and appearance descriptions of identical objects. (c) Visual object tracking: forms cycles via forward-backward trajectory consistency.

still rely on pseudo-labels (e.g., unsupervised visual tracking approaches (Zheng et al., 2021; Wang et al., 2019a; Shen et al., 2022) require initial trajectories from base trackers). To address these limitations, we propose a probabilistic modeling approach to enable universal cyclic learning across all applicable tasks.

Building upon the constraint of cycle consistency, this work establishes a unified probabilistic framework for both paired cyclic tasks (bidirectional mapping A→B and B→A) and self-cyclic tasks (A→B and B→A mapped by the same function). Methodologically, this framework is inspired by the Expectation-Maximization (EM) algorithm (Neal & Hinton, 1998), leveraging it as a foundational variational method for iterative training. By introducing a latent variable $\mathbf{z}$, it transforms the maximization of log-likelihood into the maximization of an evidence lower bound, and then approximates the optimum stepwise through the Expectation step (E-step) and Maximization step (M-step). This method has stood the test of time and remains highly influential across various fields to this day (Sun & Yang, 2020; Bao et al., 2024; Qu et al., 2019). Another canonical application is the Variational Autoencoder (VAE) (Kingma et al., 2013), which assumes that the latent variables corresponding to the data follow a prior distribution. Through variational inference, it derives a reconstruction loss and a Kullback-Leibler divergence (Kullback & Leibler, 1951) loss, ultimately training a decoder capable of generating new data samples from the prior distribution. In contrast, our framework aims to deliver theoretically rigorous and computationally efficient solutions for a broad range of cyclic learning problems.

Specifically, we formalize cycle consistency by treating intermediate data points as latent variables, with cross-task transitions as learnable distributions. Within this framework, we propose: (**i**) a universal single-step loss derived via variational inference that enforces cycle consistency for end-to-end training, and (**ii**) an EM-based method that alternately updates model parameters in two tasks when KL divergence approximation is infeasible. To validate the universality and effectiveness of our method, we conduct experiments in two distinct tasks: for image translation, our approach not only reveals the working mechanism of CycleGAN(Zhu et al., 2017) but also achieves bidirectional style mappings without GANs through the EM method. In object tracking, our model effectively captures dynamic target appearance variations via self-cyclic constraints, significantly improving unsupervised tracking robustness. The proposed probabilistic framework provides a unified solution for diverse cyclic learning scenarios. Our main contributions are:

- We regard the intermediate points (non-starting/non-terminal points) in cyclic learning as latent variables, thereby establishing the first variational probabilistic framework that unifies both paired and self-cyclic tasks through variational inference.
- We derive two theoretically grounded optimizers for general cyclic learning: (**i**) a single-step variational loss thus enabling stable and efficient training with explicit distributions, and (**ii**) a KL-free, EM-based algorithm compatible with complex distributions.
- In unpaired image translation, we theoretically explain the success of CycleGAN and propose a GAN-free, EM-based alternative. In visual tracking, we introduce CycleTrack (single-step) and CycleTrack-EM (EM-based), which achieve state-of-the-art unsupervised performance.

## 2 VARIATIONAL INFERENCE FOR CYCLIC LEARNING

### 2.1 METHODOLOGY

Fundamentally, the generation problem involves learning a function $f$ that maps data points from the input space to the output space, i.e., $f : \mathcal{X} \to \mathcal{Y}$. For example, in image captioning, $\mathcal{X}$ is a collection of natural images where a data point $\mathbf{x}$ is a photo of a horse, and $\mathcal{Y}$ is the set of all grammatically correct sentences. The corresponding $\mathbf{y} \in \mathcal{Y}$ for $\mathbf{x}$ would be a natural language description of the horse. The goal of a generative model is to learn this mapping $f$, such that for an input $\mathbf{x}$ from the domain $\mathcal{X}$, the output $f(\mathbf{x})$ appears "real" and follows a specific distribution in the codomain $\mathcal{Y}$. Although the generative function itself is not an explicit probabilistic model, it implicitly encodes the dynamic process of probabilistic transition.

We now examine a special case from a theoretical perspective: when $\mathbf{y} = f(\mathbf{x})$ is invertible. In this scenario, $f$ defines a bijective mapping, and there exists a unique inverse function $\mathbf{x} = f^{-1}(\mathbf{y})$, which is a necessary condition to guarantee cycle consistency. When a specific observed value $\hat{\mathbf{y}}$ is given, it must have been produced by a unique $\hat{\mathbf{x}} = f^{-1}(\hat{\mathbf{y}})$. Probabilistically, this implies that under the condition $\mathcal{Y} = \{\hat{\mathbf{y}}\}$, the distribution of $\mathcal{X}$ is deterministic—all probability mass is concentrated at the single point $\hat{\mathbf{x}}$. Thus, the conditional probability density function $p(\mathbf{x}|\mathbf{y})$ becomes a Dirac function:

$$p(\mathbf{x}|\mathbf{y}) = \delta(\mathbf{x} - f^{-1}(\mathbf{y})). \tag{1}$$

Let $g(\cdot)$ denote $f^{-1}(\cdot)$, with $\phi$ and $\theta$ being the parameters to be learned for $f$ and $g$ respectively, then the conditional probability can be expressed as:

$$p_\theta(\mathbf{x}|\mathbf{y}) = \delta(\mathbf{x} - g_\theta(\mathbf{y})); \quad p_\phi(\mathbf{y}|\mathbf{x}) = \delta(\mathbf{y} - f_\phi(\mathbf{x})). \tag{2}$$

Based on the above transformation relationships, cyclic learning can be formulated probabilistically to optimize the mapping functions. Specifically, for a cycle that starts from a data point $\mathbf{x}$ and returns to $\mathbf{x}$ itself, we aim to maximize the log-likelihood, i.e., $\max \log p_\theta(\mathbf{x})$. By introducing the variational distribution $q_\phi(\mathbf{y}|\mathbf{x})$, we obtain:

$$\log p_\theta(\mathbf{x}) = \int q_\phi(\mathbf{y}|\mathbf{x}) \log \frac{p_\theta(\mathbf{x}, \mathbf{y})}{p_\theta(\mathbf{y}|\mathbf{x})} d\mathbf{y} = \mathbb{E}_{q_\phi(\mathbf{y}|\mathbf{x})} \left[ \log \frac{p_\theta(\mathbf{x}, \mathbf{y})}{q_\phi(\mathbf{y}|\mathbf{x})} \right] + D_{KL}(q_\phi(\mathbf{y}|\mathbf{x})||p_\theta(\mathbf{y}|\mathbf{x})), \tag{3}$$

where $D_{KL}$ denotes the Kullback-Leibler divergence. Here, $\theta$ parameterizes only the conditional distribution $p_\theta(\mathbf{x}|\mathbf{y})$, which is usually implemented as a neural network. The joint distribution $p_\theta(\mathbf{x}, \mathbf{y})$ is defined by $p_\theta(\mathbf{x}|\mathbf{y})$ and the prior $p(\mathbf{y})$, while the posterior $p_\theta(\mathbf{y}|\mathbf{x})$ is the conditional distribution of $\mathbf{y}$ given $\mathbf{x}$ and is not directly parameterized. The first term corresponds to the so-called Evidence Lower Bound (ELBO), which admits the following decomposition:

$$\ell_{\theta,\phi}(\mathbf{x}) = \int q_\phi(\mathbf{y}|\mathbf{x}) \log p_\theta(\mathbf{x}|\mathbf{y}) d\mathbf{y} - D_{KL}(q_\phi(\mathbf{y}|\mathbf{x})||p(\mathbf{y})), \tag{4}$$

where $\int q_\phi(\mathbf{y}|\mathbf{x}) \log p_\theta(\mathbf{x}|\mathbf{y}) d\mathbf{y}$ represents the reconstruction expectation, while $D_{KL}(q_\phi(\mathbf{y}|\mathbf{x})||p(\mathbf{y}))$ enforces distributional alignment between $q_\phi(\mathbf{y}|\mathbf{x})$ and the prior $p(\mathbf{y})$. One may simultaneously consider the symmetric case starting from a data point $\mathbf{y}$ and completing the cycle back to $\mathbf{y}$, for which the ELBO is given by:

$$\ell_{\theta,\phi}(\mathbf{x}, \mathbf{y}) = \begin{aligned} &\int q_\phi(y|\mathbf{x}) \log p_\theta(\mathbf{x}|y) dy - D_{KL}(q_\phi(y|\mathbf{x})||p(y)) \\ &+ \int q_\theta(x|\mathbf{y}) \log p_\phi(\mathbf{y}|x) dx - D_{KL}(q_\theta(x|\mathbf{y})||p(x)). \end{aligned} \tag{5}$$

Here, we use $x, y$ to denote data in a generic sense, while $\mathbf{x}, \mathbf{y}$ refer to specific data instances. The gap between the maximum log-likelihood and its evidence lower bound is:

$$D_{KL}(q_\phi(\mathbf{y}|\mathbf{x})||p_\theta(\mathbf{y}|\mathbf{x})) + D_{KL}(q_\theta(\mathbf{x}|\mathbf{y})||p_\phi(\mathbf{x}|\mathbf{y})). \tag{6}$$

The maximization of $\ell_{\theta,\phi}(\mathbf{x}, \mathbf{y})$ inherently minimizes two KL divergence terms. Through this process, the variational distributions $q_\phi(\mathbf{y}|\mathbf{x})$ and $q_\theta(\mathbf{x}|\mathbf{y})$ are regularized to align with the prior distributions $p(\mathbf{y})$ and $p(\mathbf{x})$, respectively, while also being implicitly encouraged to approximate the model-derived posteriors $p_\theta(\mathbf{y}|\mathbf{x})$ and $p_\phi(\mathbf{x}|\mathbf{y})$. This dual regularization reflects the core objective of cyclic learning: to learn a pair of stochastic mappings that are jointly consistent with both the marginal structure of the data and the cyclic reconstruction constraint, thereby forming a probabilistically coherent bidirectional translation between the two domains.

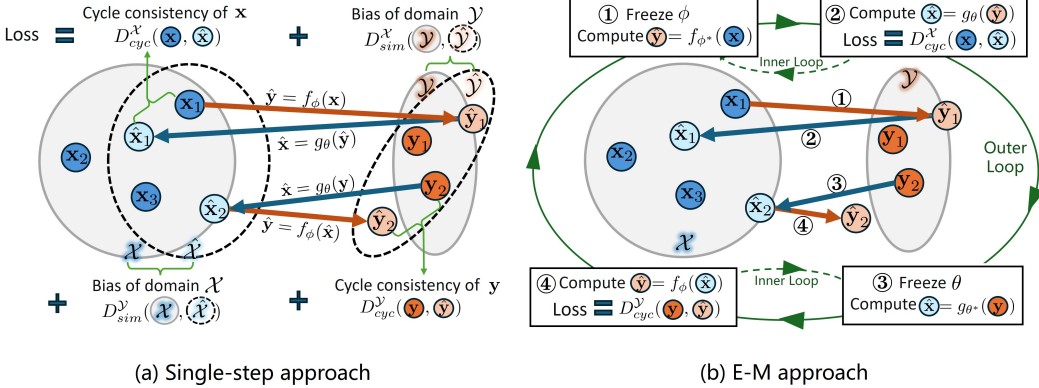

(a) Single-step approach                    (b) E-M approach

Figure 2: General solution for cyclic learning. (a) Single-step joint optimization. (b) Dual EM-iteration optimization.

This probabilistic formulation shares conceptual roots with both variational autoencoders (Kingma et al., 2013) and the expectation–maximization algorithm (Neal & Hinton, 1998). In fact, the two specific learning strategies we introduce next can be seen as their direct analogues: one follows a VAE-style joint optimization, and the other adopts an EM-style iterative optimization. What distinguishes our framework from the previous methods is the role of the latent variable $\mathbf{y}$. Rather than being a free latent variable used solely to model the distribution of $\mathbf{x}$, here $\mathbf{y}$ is a real observed variable from domain $\mathcal{Y}$. It carries its own structure and constraints, and must satisfy the marginal prior $p(\mathbf{y})$ estimated from data. The learning objective is not merely to reconstruct $\mathbf{x}$, but to learn bidirectional mappings that are probabilistically consistent with both domains, forming a cycle that respects the intrinsic properties of $\mathbf{y}$.

Returning to the perspective of mapping functions, for the first term in Eq. 4, we have:

$$\int q_\phi(\mathbf{y}|\mathbf{x})\log p_\theta(\mathbf{x}|\mathbf{y})d\mathbf{y} = \int \delta(\mathbf{y}-f_\phi(\mathbf{x}))\log\delta(\mathbf{x}-g_\theta(\mathbf{y}))d\mathbf{y} = \log\delta(\mathbf{x}, g_\theta(f_\phi(\mathbf{x}))). \quad (7)$$

Then, for a distance function $D_{cyc}(\mathbf{x},\hat{\mathbf{x}})$ that is zero iff $\mathbf{x}=\hat{\mathbf{x}}$ and positive otherwise, maximizing the expected log-likelihood reduces to minimizing $D_{cyc}$:

$$\arg\max_{\theta,\phi}\mathbb{E}_{q_\phi(\mathbf{y}|\mathbf{x})}\left[\log p_\theta(\mathbf{x}|\mathbf{y})\right] = \arg\max_{\theta,\phi}\log\delta(\mathbf{x}, g_\theta(f_\phi(\mathbf{x}))) = \arg\min_{\theta,\phi}D_{cyc}\left(\mathbf{x}, g_\theta\left(f_\phi(\mathbf{x})\right)\right). \quad (8)$$

Here the cycle-consistency loss $D_{cyc}$ serves as a task-specific proxy for the probabilistic expectation, directly minimizing the discrepancy between $g_\theta(f_\phi(\mathbf{x}))$ and $\mathbf{x}$.

Under the deterministic mapping assumption, where $q_\phi(\mathbf{y}|\mathbf{x})$ collapses to a Dirac delta $\delta(\mathbf{y}-f_\phi(\mathbf{x}))$, the second term in Eq. 4 reduces to

$$D_{KL}(q_\phi(\mathbf{y}|\mathbf{x})||p(\mathbf{y})) = \int\delta(\mathbf{y}-f_\phi(\mathbf{x}))\cdot log\frac{\delta(\mathbf{y}-f_\phi(\mathbf{x}))}{p(\mathbf{y})}d\mathbf{y} = -\log p(f_\phi(\mathbf{x})) + \text{constant}. \quad (9)$$

where the constant arises from the ill-defined $\log\delta(0)$ and can be ignored in optimization. Thus, maximizing the negative KL divergence is equivalent to maximizing $\log p(f_\phi(\mathbf{x}))$, i.e., encouraging the mapped output $f_\phi(\mathbf{x})$ to lie in regions where the prior $p(\mathbf{y})$ is high.

In practice, $p(\mathbf{y})$ may not be explicitly modeled, especially when $\mathbf{y}$ resides in a complex domain. In such cases, we can replace $\log p(f_\phi(\mathbf{x}))$ with a distance function $D_{sim}(f_\phi(\mathbf{x}),\mathcal{Y})$ that measures how well the generated sample fits the target domain $\mathcal{Y}$. This leads to the following equivalence:

$$\arg\max_\phi - D_{KL}(q_\phi(\mathbf{y}|\mathbf{x})||p(\mathbf{y})) = \arg\max_\phi\log p(f_\phi(\mathbf{x})) = \arg\min_\phi D_{sim}(f_\phi(\mathbf{x}),\mathcal{Y}). \quad (10)$$

A classic choice for $D_{sim}$ is the Wasserstein distance (Vaserstein, 1969), which provides a meaningful geometric discrepancy measure. Combining Eq. 4 with Eq. 8 and Eq. 10, the overall optimization objective for the deterministic cyclic mapping can be approximated as

$$\arg\max_{\theta,\phi}\ell_{\theta,\phi}(\mathbf{x}) \approx \arg\min_{\theta,\phi}\left(D_{cyc}(\mathbf{x}, g_\theta(f_\phi(\mathbf{x}))) + D_{sim}(f_\phi(\mathbf{x}),\mathcal{Y})\right). \quad (11)$$

Note that $\ell_{\theta,\phi}(\mathbf{x})$ requires joint optimization of $\theta$ and $\phi$, which may not achieve exact equality in Eq. 11 due to differing gradient behaviors across distance metrics. Nevertheless, since both $D_{cyc}$ and $D_{sim}$ are proxy methods, minimizing $D_{cyc} + D_{sim}$ to approximate $\max \ell_{\theta,\phi}(\mathbf{x})$ is not detrimental to training. The core design principle requires the loss function to incorporate:

- The similarity between $\hat{\mathbf{x}}$ and $\mathbf{x}$ (measured by $D_{cyc}$)
- The degree to which $\hat{\mathbf{y}}$ belongs to $\mathcal{Y}$ (measured by $D_{sim}$)

For cyclic tasks, models may converge to local optima of either $\min D_{cyc}$ or $\min D_{sim}$. This necessitates task-specific balancing between these two approximating terms.

By integrating Eq. 5 with Eq. 11, we arrive at the two-way cycle-consistent loss:

$$\mathcal{L}(\mathbf{x}, \mathbf{y}) = D_{cyc}^{\mathcal{X}}(\mathbf{x}, g_\theta(f_\phi(\mathbf{x}))) + D_{sim}^{\mathcal{X}}(f_\phi(\mathbf{x}), \mathcal{Y}) + D_{cyc}^{\mathcal{Y}}(\mathbf{y}, f_\phi(g_\theta(\mathbf{y}))) + D_{sim}^{\mathcal{Y}}(g_\theta(\mathbf{y}), \mathcal{X}), \quad (12)$$

which corresponds to the direct optimization scheme shown in Fig. 2(a).

The variational interpretation in Eq. 5 suggests a natural alternating optimization strategy. By treating the two mappings as coupled latent-variable models, we can view the learning process as jointly maximizing $\log p_\theta(\mathbf{x})$ and $\log p_\phi(\mathbf{y})$, where each benefits from the other's intermediate outputs. This motivates an EM-based procedure that iterates between two complementary stages: fixing one mapping to infer pseudo-labels for the other.

Concretely, as illustrated in Fig. 2(b), the $E_\theta$-$M_\theta$ stage first uses the current forward mapping $f_\phi$ to generate $\hat{\mathbf{y}} = f_\phi(\mathbf{x})$ from a sampled batch of $\mathbf{x}$ (E-step), then updates the backward mapping $g_\theta$ by minimizing the cycle loss $D_{cyc}(\mathbf{x}, g_\theta(\hat{\mathbf{y}}))$ (M-step). Symmetrically, the $E_\phi$-$M_\phi$ stage samples $\mathbf{y}$, produces $\hat{\mathbf{x}} = g_\theta(\mathbf{y})$, and updates $f_\phi$ via $D_{cyc}(\mathbf{y}, f_\phi(\hat{\mathbf{x}}))$. This alternating scheme promotes stable convergence by progressively aligning both mappings with the cyclic consistency constraint.

From a probabilistic perspective, in the $E_\theta$-$M_\theta$ stage, we assume the current forward mapping $f_\phi$ is sufficiently accurate, i.e., $q_\phi(\mathbf{y}|\mathbf{x})$ approximates the true posterior $p(\mathbf{y}|\mathbf{x})$. Under this assumption, the $E_\theta$ step (sampling $\mathbf{x}$ and generating $\hat{\mathbf{y}} = f\phi(\mathbf{x})$) effectively enforces $D_{\mathrm{KL}}(q_\phi(\mathbf{y}|\mathbf{x})|p(\mathbf{y}|\mathbf{x})) = 0$, which implies $q_\phi(\mathbf{y}|\mathbf{x}) = p(\mathbf{y}|\mathbf{x})$. The subsequent $M_\theta$ step then updates $\theta$ by maximizing $\mathbb{E}_{q_\phi(\mathbf{y}|\mathbf{x})}[\log p_\theta(\mathbf{x}|\mathbf{y})]$, driving $\theta$ toward a lower bound determined by the approximation quality of $q_\phi$.

---

**Algorithm 1** An EM approach for cycle-consistent tasks.

**Input:** Dataset $\mathcal{X} = \{\mathbf{x}^i\}_{i=1}^N$, $\mathcal{Y} = \{\mathbf{y}^i\}_{i=1}^M$
1: **while** not converge **do**
2:      **while** insufficient loss decrease **do**
3:          Sample batch of datapoints $\mathcal{X}' = \{\mathbf{x}\}$ from $\mathcal{X}$
4:          Get pseudo-labels $\hat{\mathbf{y}} = f_\phi(\mathbf{x})$ for each $\mathbf{x} \in \mathcal{X}'$ $\backslash\backslash$ $E_\theta$
5:          Update $\theta$ via $\mathcal{L}(\theta) = D_{cyc}^{\mathcal{X}}(\mathbf{x}, g_\theta(\hat{\mathbf{y}}))$ $\backslash\backslash$ $M_\theta$
6:      **end while**
7:      **while** insufficient loss decrease **do**
8:          Sample batch of datapoints $\mathcal{Y}' = \{\mathbf{y}\}$ from $\mathcal{Y}$
9:          Get pseudo-labels $\hat{\mathbf{x}} = g_\theta(\mathbf{y})$ for each $\mathbf{y} \in \mathcal{Y}'$ $\backslash\backslash$ $E_\phi$
10:         Update $\phi$ via $\mathcal{L}(\phi) = D_{cyc}^{\mathcal{Y}}(\mathbf{y}, f_\phi(\hat{\mathbf{x}}))$ $\backslash\backslash$ $M_\phi$
11:      **end while**
12: **end while**
**Output:** Generative models $g_\theta(\cdot)$ and $f_\phi(\cdot)$.

---

Symmetrically, in the $E_\phi$-$M_\phi$ stage, we assume $g_\theta$ is sufficiently accurate, such that $p_\theta(\mathbf{x}|\mathbf{y})$ effectively represents the true conditional distribution $p(\mathbf{x}|\mathbf{y})$. The $E_\phi$ step (sampling $\mathbf{y}$ and generating $\hat{\mathbf{x}} = g_\theta(\mathbf{y})$) then enforces $D_{\mathrm{KL}}(q_\theta(\mathbf{x}|\mathbf{y})|p(\mathbf{x}|\mathbf{y})) = 0$, and the $M_\phi$ step updates $\phi$ by maximizing $\mathbb{E}_{q_\theta(\mathbf{x}|\mathbf{y})}[\log p_\phi(\mathbf{y}|\mathbf{x})]$. Algorithm 1 summarizes the complete procedure, but with the tone of generative models.

In the standard EM algorithm, the E-step fixes parameters $\theta$ to find the latent distribution $q(\mathbf{z})$ that tightens the ELBO, typically by computing the posterior $p(\mathbf{z}|\mathbf{x}, \theta)$. The M-step then fixes $q(\mathbf{z})$ and updates $\theta$ to maximize the expected complete-data log-likelihood. This process can be viewed as coordinate ascent, alternately optimizing $\theta$ and $q(\mathbf{z})$. Our method employs a different variational inference workflow since we focus on learning mappings $p(\mathbf{x}|\mathbf{y})$ and $p(\mathbf{y}|\mathbf{x})$ rather than data distributions. When optimizing $p_\theta(\mathbf{x}|\mathbf{y})$, the E-step fixes $\theta$ and finds the optimal $q_\phi(\mathbf{y}|\mathbf{x})$ — this is accomplished by the $E_\phi$-$M_\phi$ stage, which updates $\phi$ to better approximate the posterior. The M-step then maximizes the expected log-likelihood under this $q_\phi(\mathbf{y}|\mathbf{x})$ — corresponding to the $E_\theta$-$M_\theta$ stage, which updates $\theta$ using the generated pseudo-labels. The same logic applies to $p_\phi(\mathbf{y}|\mathbf{x})$. Parameters $\theta$ and $\phi$ are alternately updated while keeping the other fixed. For clarity, we define fixing network

Table 1: The correspondence between components in Eq.12 and those in CycleGAN.

Table 2: The correspondence between steps in Algo. 1 and those in CycleGN.

| Components in Eq. 12 | Components in CycleGAN |
|---|---|
| $D_{cyc}^{\mathcal{X}}(\mathbf{x}, g_\theta(f_\phi(\mathbf{x})))$ | $\mathbb{E}_{\mathbf{x}\sim p_{data}(\mathbf{x})}[\|g_\theta(f_\phi(\mathbf{x})) - \mathbf{x}\|_1]$ |
| $D_{sim}^{\mathcal{X}}(f_\phi(\mathbf{x}), \mathcal{Y})$ | $\mathcal{L}_{\text{GAN}}(f_\phi, D_{\mathcal{Y}}, \mathcal{X}, \mathcal{Y})$ |
| $D_{cyc}^{\mathcal{Y}}(\mathbf{y}, f_\phi(g_\theta(\mathbf{y})))$ | $\mathbb{E}_{\mathbf{y}\sim p_{data}(\mathbf{y})}[\|f_\phi(g_\theta(\mathbf{y})) - \mathbf{y}\|_1]$ |
| $D_{sim}^{\mathcal{Y}}(g_\theta(\mathbf{y}), \mathcal{X})$ | $\mathcal{L}_{\text{GAN}}(g_\theta, D_{\mathcal{X}}, \mathcal{Y}, \mathcal{X})$ |

| Steps in Algo. 1. | Steps in CycleGN |
|---|---|
| $E_\theta$ | $\hat{\mathbf{y}} = f_\phi(\mathbf{x})$ |
| $M_\theta$ | Update $\theta$ via $\mathcal{L}_{\text{cyc}}(g_\theta) = \mathbb{E}_{\mathbf{x}\sim p_{data}(\mathbf{x})}[\|g_\theta(\hat{\mathbf{y}}) - \mathbf{x}\|_1]$ |
| $E_\phi$ | $\hat{\mathbf{x}} = g_\theta(\mathbf{y})$ |
| $M_\phi$ | Update $\phi$ via $\mathcal{L}_{\text{cyc}}(f_\phi) = \mathbb{E}_{\mathbf{y}\sim p_{data}(\mathbf{y})}[\|f_\phi(\hat{\mathbf{x}}) - \mathbf{y}\|_1]$ |

parameters and sampling data as the E-step, and maximizing distribution functions as the M-step, labeled as $E_\phi$-$M_\phi$ and $E_\theta$-$M_\theta$ based on the optimized parameters. Essentially, $E_\phi$-$M_\phi$ serves as the E-step for $p_\theta(\mathbf{x}|\mathbf{y})$ while $E_\theta$-$M_\theta$ acts as its M-step. Thus, our method also implements coordinate ascent, alternately optimizing model parameters and parameterized distribution functions.

This approach eliminates the need to define $D_{sim}$ as in Eq. 12, thus avoiding both training instability caused by metric inaccuracies or ill-defined data distributions. A key question then arises: how does the EM iteration ensure that $\hat{\mathbf{y}} = f_\phi(\mathbf{x}) \in \mathcal{Y}$? The answer stems from its alternating optimization nature: in the second M-step, we enforce $\hat{\mathbf{y}} = f_\phi(g_\theta(\mathbf{y}))$ to approximate $\mathbf{y}$, thereby guaranteeing that the output of $f_\phi(\cdot)$ falls within $\mathcal{Y}$. By symmetry, this also ensures $\hat{\mathbf{x}} = g_\theta(\mathbf{y}) \in \mathcal{X}$.

## 2.2 APPLICATION ON UNPAIRED IMAGE TRANSLATION

We use CycleGAN(Zhu et al., 2017) as an example and conduct experiments on the unpaired image-to-image translation task. For two distinct image domains $\mathcal{X}$ and $\mathcal{Y}$, the objective of CycleGAN is to find a pair of mapping functions, denoted as $f_\phi : \mathcal{X} \rightarrow \mathcal{Y}$ and $g_\theta : \mathcal{Y} \rightarrow \mathcal{X}$. The method employs a end-to-end training strategy for network training, with the proposed loss function as follows:

$$\mathcal{L}(f_\phi, g_\theta, D_{\mathcal{X}}, D_{\mathcal{Y}}) = \mathcal{L}_{\text{GAN}}(f_\phi, D_{\mathcal{Y}}, \mathcal{X}, \mathcal{Y}) + \mathcal{L}_{\text{GAN}}(g_\theta, D_{\mathcal{X}}, \mathcal{Y}, \mathcal{X}) + \mathcal{L}_{\text{cyc}}(f_\phi, g_\theta), \quad (13)$$

where $\mathcal{L}_{\text{GAN}}$ denotes the adversarial loss and $\mathcal{L}_{\text{cyc}}$ represents the cycle consistency loss. $D_{\mathcal{X}}$ and $D_{\mathcal{Y}}$ are the discriminators for domains $\mathcal{X}$ and $\mathcal{Y}$ respectively, which engage in adversarial training with the generators $g_\theta$ and $f_\phi$. $\mathcal{L}_{\text{cyc}}$ consists of both forward and backward cycle consistency losses:

$$\mathcal{L}_{\text{cyc}}(f_\phi, g_\theta) = \mathbb{E}_{\mathbf{x}\sim p_{data}(\mathbf{x})}[\|g_\theta(f_\phi(\mathbf{x})) - \mathbf{x}\|_1] + \mathbb{E}_{\mathbf{y}\sim p_{data}(\mathbf{y})}[\|f_\phi(g_\theta(\mathbf{y})) - \mathbf{y}\|_1]. \quad (14)$$

Under our framework, the corresponding components of the loss function can be mapped onto those in Eq.12, as summarized in Tab. 1. $\mathcal{L}_{\text{GAN}}$ encourages the generated distribution to resemble the target distribution, while $\mathcal{L}_{\text{cyc}}$ ensures cycle consistency. The adversarial training of a GAN discriminator is intrinsically linked to minimizing the Jensen-Shannon (JS) divergence (Lin, 2002) between the generated and real data distributions (Goodfellow et al., 2014). As the JS divergence is a symmetric variant of the KL divergence, the adversarial loss can be viewed as an indirect means of optimizing a divergence related to $D_{KL}$. By integrating these components, it becomes evident that Eq. 13 is a specific instantiation of our general formulation in Eq. 12. This connection provides a theoretical explanation for the effectiveness of CycleGAN.

Based on Algo. 1, we propose a cyclic learning approach that alternately optimizes the forward and backward tasks. The detailed procedure is summarized in Tab. 2, with iterations continuing until convergence is achieved. Unlike CycleGAN, our method eliminates adversarial discriminators entirely and is thus named CycleGN.

## 2.3 EXPERIMENTS

CycleGN and CycleGAN(Zhu et al., 2017) adopt the same generator architecture as pix2pix (Isola et al., 2017). The unpaired training and test images are sourced from the Cityscapes dataset (Cordts et al., 2016). CycleGN alternates between training $E_\theta$-$M_\theta$ and $E_\phi$-$M_\phi$ every 200 samples, with a total of 100 training epochs. All other training configurations are identical to CycleGAN. We compare our approach against several methods that employ different loss functions for cyclic learning, including CoGAN (Liu & Tuzel, 2016), BiGAN/ALI (Dumoulin et al., 2016; Donahue et al., 2016), SimGAN (Shrivastava et al., 2017), and the feature loss combined with GAN (Shrivastava et al., 2017; Zhu et al., 2017). We evaluate our method on the Cityscapes dataset for both labels-to-photo

Table 3: FCN-scores of labels→photo for different methods on Cityscapes.

| Loss | GAN | Per-pixel acc. | Per-class acc. | Class IOU |
|---|---|---|---|---|
| CoGAN | ✓ | 0.40 | 0.10 | 0.06 |
| BiGAN/ALI | ✓ | 0.19 | 0.06 | 0.02 |
| SimGAN | ✓ | 0.20 | 0.10 | 0.04 |
| Feat. loss + GAN | ✓ | 0.06 | 0.04 | 0.01 |
| CycleGAN | ✓ | **0.52** | **0.17** | **0.11** |
| CycleGN (ours) | ✗ | **0.52** | 0.14 | 0.10 |

Table 4: Classification performance of photo→labels on Cityscapes.

| Loss | GAN | Per-pixel acc. | Per-class acc. | Class IOU |
|---|---|---|---|---|
| CoGAN | ✓ | 0.45 | 0.11 | 0.08 |
| BiGAN/ALI | ✓ | 0.41 | 0.13 | 0.07 |
| SimGAN | ✓ | 0.47 | 0.11 | 0.07 |
| Feat. loss + GAN | ✓ | 0.50 | 0.10 | 0.06 |
| CycleGAN | ✓ | **0.58** | **0.22** | **0.16** |
| CycleGN (ours) | ✗ | 0.51 | 0.16 | 0.10 |

and photo-to-labels translation tasks. Tab. 3 and 4 compare CycleGN with CycleGAN and other loss configurations. The successes achieved by CycleGAN on this pair of cyclic tasks demonstrate the feasibility of the single-step optimization paradigm. Moreover, our proposed application of CycleGN based on the EM method achieves better accuracy than other loss functions, trailing only marginally behind CycleGAN. Notably, CycleGN achieves competitive generation results without any adversarial structure, simply by pushing the generative network's outputs closer to target domain instances.

# 3 EXTENDING TO SELF-CYCLIC LEARNING

## 3.1 METHODOLOGY

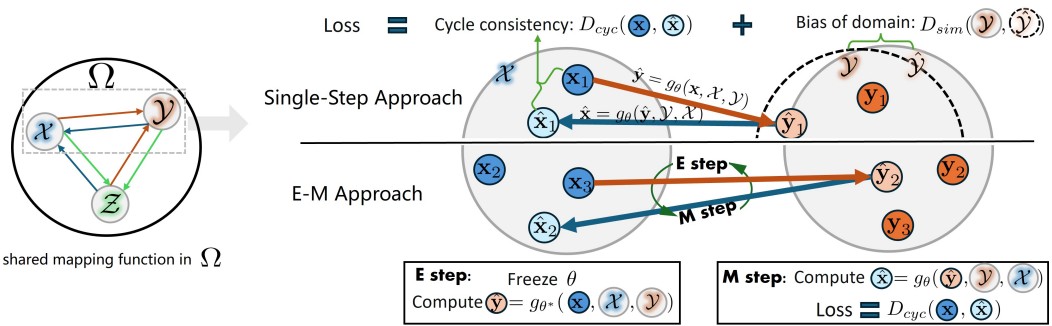

Figure 3: General solution for self-cyclic learning.

We consider a special case with single-task learning as illustrated in Fig. 3, where for any $\mathcal{X}, \mathcal{Y} \in \Omega$, and any $\mathbf{x} \in \mathcal{X}$, $\mathbf{y} \in \mathcal{Y}$, the symmetry $\theta = \phi$ holds. When $f_\phi = g_\theta$, a trivial solution $g_\theta(g_\theta(\mathbf{x})) = \mathbf{x}$ would be $g_\theta(\mathbf{x}) = \mathbf{x}$. However, since $\hat{\mathbf{y}} = g_\theta(\mathbf{x})$ must belong to $\mathcal{Y}$ not $\mathcal{X}$ in cyclic learning, $g_\theta(\mathbf{x})$ cannot directly equal $\mathbf{x}$. Thus, we reformulate the optimization objective of cyclical learning as $g_\theta(g_\theta(\mathbf{x}, \mathcal{X}, \mathcal{Y}), \mathcal{Y}, \mathcal{X}) = \mathbf{x}$, which corresponds to optimizing the

---

**Algorithm 2** An EM approach for self-cyclic tasks.

**Input:** Dataset $\Omega = \{\mathcal{X}^i\}_{i=1}^{\mathbf{N}}$ with $\mathcal{X}^i = \{\mathbf{x}^{ij}\}_{j=1}^{N^i}$
1: **while** not converge **do**
2:     Random chosen sub-domains $\mathcal{X}, \mathcal{Y} \subseteq \Omega$
3:     Sample batch of datapoints $\mathcal{X}' = \{\mathbf{x}\}$ from $\mathcal{X}$
4:     ⊡ **E-step: Stop Gradient**
5:     Get pseudo-labels $\hat{\mathbf{y}} = g_\theta(\mathbf{x}, \mathcal{X}, \mathcal{Y})$ for each $x \in \mathcal{X}'$
6:     ⊡ **M-step: Learning Procedure**
7:     Update $\theta$ via $\mathcal{L}(\theta) = D_{cyc}(\mathbf{x}, g_\theta(\hat{\mathbf{y}}, \mathcal{Y}, \mathcal{X}))$
8: **end while**
**Output:** Generative model $g_\theta(\cdot)$.

---

conditional probability $\log p_\theta(\mathbf{x}|\mathcal{Y}, \mathcal{X})$, yielding the evidence lower bound:

$$\ell_\theta(\mathbf{x}|\mathcal{Y}, \mathcal{X}) = \mathbb{E}_{q_\theta(\mathbf{y}|\mathbf{x}, \mathcal{X}, \mathcal{Y})}\left[\log p_\theta(\mathbf{x}|\mathbf{y}, \mathcal{Y}, \mathcal{X})\right] - D_{KL}(q_\theta(\mathbf{y}|\mathbf{x}, \mathcal{X}, \mathcal{Y})||p(\mathbf{y})). \quad (15)$$

For self-cyclic tasks, either the backward or forward components in the asymmetric task loss in Eq. 12 can be removed, leading to the following loss function:

$$\mathcal{L}(\mathbf{x}) = D_{cyc}(\mathbf{x}, g_\theta(g_\theta(\mathbf{x}, \mathcal{X}, \mathcal{Y}), \mathcal{Y}, \mathcal{X})) + D_{sim}(g_\theta(\mathbf{x}, \mathcal{X}, \mathcal{Y}), \mathcal{Y}). \quad (16)$$

This loss function guides the model $\hat{\mathbf{y}} = g_\theta(\cdot, \cdot, \mathcal{Y})$ toward $\hat{\mathbf{y}} \in \mathcal{Y}$ through $D_{sim}$, preventing convergence to the local optimum $g_\theta(\mathbf{x}, \mathcal{X}, \cdot) = \mathbf{x}$.

The EM algorithm remains applicable to self-cyclic learning. As outlined in Algo. 2, the E-step minimizes $D_{KL}(q_\theta(\mathbf{y}|\mathbf{x}, \mathcal{X}, \mathcal{Y})||p_{\theta*}(\mathbf{y}|\mathbf{x}, \mathcal{X}, \mathcal{Y}))$, while the M-step maximizes the expected log-likelihood $\arg\max \mathbb{E}_{p_{\theta*}(\mathbf{y}|\mathbf{x}, \mathcal{X}, \mathcal{Y})}[\log p_\theta(\mathbf{x}|\mathbf{y}, \mathcal{Y}, \mathcal{X})]$. The KL divergence term in Eq. 15 is omitted in M-step as it reduces to a constant in this case. Note that $f_\phi = g_\theta$ enables joint optimization of bidirectional tasks in one EM process, contrasting with the $E_\theta$-$M_\theta$ / $E_\phi$-$M_\phi$ alternating sequence in Algo. 1.

## 3.2 Application on Unsupervised Visual Tracking

Visual object tracking is a classic self-cycle-consistent task, with its cyclic structure illustrated in Fig. 1(c). Based on the paradigm proposed in this work, we can design two self-supervised schemes.

For a video sequence, let $\mathcal{X}$ and $\mathcal{Y}$ be the selected template and search frames, with $\mathbf{x}$ as a random object box in $\mathcal{X}$. Then, the loss function of the tracker $T$ can be derived from Eq. 16:

$$\mathcal{L}(T, \mathbf{x}) = \mathcal{L}_b(\mathbf{x}, T(T(\mathbf{x}, \mathcal{X}, \mathcal{Y}), \mathcal{Y}, \mathcal{X})) + \mathcal{L}_b(T(\mathbf{x}, \mathcal{X}, \mathcal{Y}), \tilde{\mathbf{y}}),$$
$$s.t. \quad \tilde{\mathbf{y}} = \arg\max_{\mathbf{y} \in \text{BOX}_\mathcal{Y}} \text{IoU}(\mathbf{y}, T(\mathbf{x}, \mathcal{X}, \mathcal{Y})), \tag{17}$$

where $\mathcal{L}_b$ is the bounding-box loss, IoU represents the Intersection over Union between two bounding boxes, and $\text{BOX}_\mathcal{Y}$ is detector-generated bounding-box set in frame $\mathcal{Y}$. The correspondence between this loss function and Eq.16 is as follows:

- $D_{cyc}(\mathbf{x}, g_\theta(g_\theta(\mathbf{x}, \mathcal{X}, \mathcal{Y}), \mathcal{Y}, \mathcal{X})) \Rightarrow \mathcal{L}_b(\mathbf{x}, T(T(\mathbf{x}, \mathcal{X}, \mathcal{Y}), \mathcal{Y}, \mathcal{X}))$;
- $D_{sim}(g_\theta(\mathbf{x}, \mathcal{X}, \mathcal{Y}), \mathcal{Y}) \Rightarrow \mathcal{L}_b(T(\mathbf{x}, \mathcal{X}, \mathcal{Y}), \tilde{\mathbf{y}})$.

Using Algo. 2 as a reference, we also propose an EM variant that bypasses $\mathbf{y}$ distribution estimation, where the expectation step computes $\hat{\mathbf{y}} = T(\mathbf{x}, \mathcal{X}, \mathcal{Y})$, and the maximization step updates $T$ by optimizing the objective $\mathcal{L}_b(\mathbf{x}, T(\hat{\mathbf{y}}, \mathcal{Y}, \mathcal{X}))$.

Since current trackers fail to achieve differentiable head-to-tail connections, we are compelled to develop CycleTrack from scratch. We map template boxes $\mathbf{x}$ to positional tokens via MLP, then concatenate them with uncropped frame tokens as input. The main architecture combines a vanilla ViT (Dosovitskiy et al., 2021) encoder, STARK's feature enhancer (Yan et al., 2021), and parallel FCOS heads (Tian et al., 2019) that generate confidence-weighted box outputs. The tracker can be formally expressed as $\hat{\mathbf{y}} = T_\theta(\mathbf{x}, \mathcal{X}, \mathcal{Y})$. For clarity, we refer to the single-step trained tracker as CycleTrack and the EM-trained tracker as CycleTrack-EM.

## 3.3 Experiments

Table 5: Comparison with leading unsupervised trackers on LaSOT and TrackingNet.

| Method | LaSOT | | TrackingNet | |
|---|---|---|---|---|
| | AUC | Precision | AUC | Precision |
| ResPUL | - | - | 54.6 | 48.5 |
| LUDT+ | 30.5 | 28.8 | 56.3 | 49.5 |
| USOT* | 35.8 | 34.0 | 61.5 | 56.6 |
| ULAST*-off | 46.8 | 44.8 | 64.9 | 58.5 |
| ULAST*-on | 47.1 | 45.1 | 65.4 | 59.2 |
| CycleTrack | _51.0_ | _49.7_ | _75.9_ | _71.5_ |
| CycleTrack-EM | **56.5** | **57.9** | **77.3** | **74.4** |

Table 6: Comparison with leading strictly-unsupervised trackers on two datasets.

| Method | LaSOT | | TrackingNet | |
|---|---|---|---|---|
| | AUC | Precision | AUC | Precision |
| ResPUL | - | - | 54.6 | 48.5 |
| LUDT | 26.2 | 23.4 | 54.3 | 46.9 |
| USOT | 33.7 | 32.3 | 59.9 | 55.1 |
| ULAST-off | 42.9 | 40.5 | - | - |
| ULAST-on | 43.3 | 40.7 | - | - |
| CycleTrack | _45.0_ | _42.2_ | _65.6_ | _59.0_ |
| CycleTrack-EM | **51.2** | **49.9** | **69.1** | **64.7** |

In CycleTrack and CycleTrack-EM, $\mathcal{L}_b$ adopts the weighted L1 and GIoU(Rezatofighi et al., 2019) losses consistent with STARK(Yan et al., 2021). The DETA detector(Ouyang-Zhang et al., 2022) produces detections on TrackingNet(Muller et al., 2018), GOT-10k(Huang et al., 2019), and La-SOT(Fan et al., 2019), combined with COCO(Lin et al., 2014) for unsupervised training. Optical flow labels are generated by ARFlow(Liu et al., 2020) on YouTube-VOS(Xu et al., 2018), ImageNet-ViD(Deng et al., 2009), GOT-10k, and LaSOT, serving as the strictly unsupervised set. We conduct

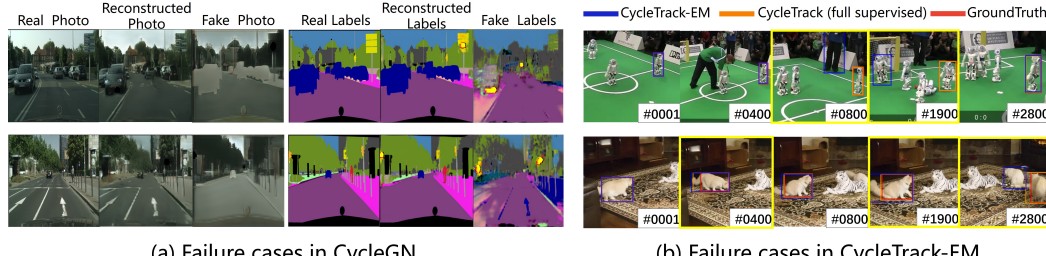

(a) Failure cases in CycleGN

(b) Failure cases in CycleTrack-EM

Figure 4: Failure cases in cyclic learning. (a) CycleGN failures in photo↔map translation: While reconstructed images closely resemble real images, the generated fake images exhibit significant quality degradation. (b) Mislocalizations in CycleTrack-EM (highlighted by yellow boxes): Target objects are confused with other salient objects or visually similar distractors.

comparisons with leading unsupervised trackers, including ResPUL(Wu et al., 2021), LUDT(Wang et al., 2021), USOT(Zheng et al., 2021), ULAST(Shen et al., 2022). We evaluate our unsupervised object tracking methods on LaSOT and TrackingNet, demonstrating significant advantages over existing approaches. Unlike conventional methods requiring pseudo-labels for image cropping, our tracker directly implements the proposed cyclic learning paradigm through full-image processing. Experimental results in Tab. 5 and 6 show our single-step and EM-based training approaches achieve state-of-the-art performance in both unsupervised (detector-annotated) and strictly unsupervised (optical-flow-annotated) settings, outperforming the second-best methods by considerable margins. Additionally, our framework naturally supports semi-supervised training (see Appendix for more details).

## 4 FURTHER ANALYSIS AND DISCUSSION

This section outlines the primary concerns of cyclic learning in applied contexts.

**1. Is the mapping learned by cyclic learning always what we need?**

Not necessarily. We observe that cyclic learning models can learn incorrect mappings across paired domains. As shown in Fig. 4, although the reconstructed photos can closely match the real photos (and similarly for the reconstructed maps and the real maps), the intermediate fake images remain unsatisfactory. In visual tracking, CycleTrack occasionally locates incorrect objects in search frames. Owing to the annotation bias of the detector, the tracker can still probabilistically re-localize the target even when initialized with random objects. This may cause the tracker to degenerate into a basic object detector.

We attribute these failure cases primarily to the inherent limitations of cyclic learning itself. Without paired annotations, the models only learn some mapping between domains $\mathcal{X}$ and $\mathcal{Y}$ that satisfies cycle consistency but does not guarantee that the learned mapping is exactly what we intend. Introducing additional constraints beyond cycle consistency may help mitigate this issue.

Additionally, the EM method has an intrinsic risk of converging to local optima. During training, $g_\theta$ may gradually adapt to two distinct modes: $g_\theta(\mathbf{a}) = \mathbf{x}$ and $g_\theta(\mathbf{y}) = \mathbf{b}$. This means it simultaneously learns mappings from $\mathcal{A} \to \mathcal{X}$ and $\mathcal{Y} \to \mathcal{B}$, both of which can be achieved with a single set of parameters $\theta$, where $\mathcal{A} = \{\mathbf{a}^i\}$ and $\mathcal{B} = \{\mathbf{b}^i\}$ represent domains that emerge as local optima. Correspondingly, $f_\phi$ learns the mappings $\mathcal{X} \to \mathcal{A}$ and $\mathcal{B} \to \mathcal{Y}$. As a result, we obtain a pair of bidirectional mappings: $\mathcal{X} \leftrightarrow \mathcal{A}$ and $\mathcal{B} \leftrightarrow \mathcal{Y}$, which deviate from the intended behavior $\mathcal{X} \leftrightarrow \mathcal{Y}$. Once a steady state where both cycles are satisfied is reached, the assumption $p(\mathbf{y}|\mathbf{x}) = q_\phi(\mathbf{y}|\mathbf{x})$ in an inner loop can never be achieved.

**2. Which is better - single-step training or EM training?**

In image translation, the single-step CycleGAN outperforms the EM-based CycleGN. Conversely, in visual tracking, the EM-based method CycleTrack-EM surpasses its single-step loss variant, CycleTrack. Besides, as introduced in the opening section, image generation tasks(Kwon & Park, 2019;

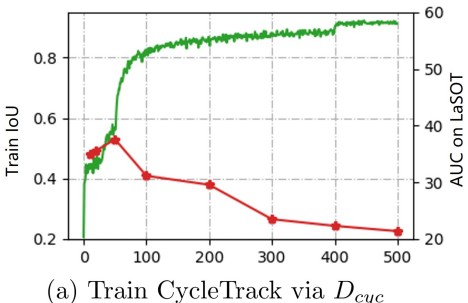 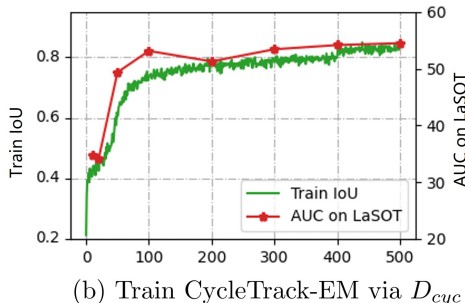

(a) Train CycleTrack via $D_{cyc}$      (b) Train CycleTrack-EM via $D_{cyc}$

Figure 5: Ablation on $D_{sim}$ removal (E-step removal): (a) Train tracker solely with $D_{cyc}$; (b) Freeze forward process via E-step, then train tracker with $D_{cyc}$.

Yang et al., 2020) tend to favor single-step optimization, whereas the REC-REG cycle(Yue et al., 2024; Wang et al., 2024) exhibits a preference for EM optimization. We argue that the choice between these two methods primarily depends on whether $D_{KL}$ is well approximated. Both methods must employ the cycle-consistency loss $D_{cyc}$, which can be disregarded in this comparison. The accuracy of $D_{sim}$ in measuring the KL divergence between sample and target distributions critically influences convergence behavior. For instance, while EM methods may introduce training instability, the GAN discriminator provides a robust $D_{KL}$ surrogate that makes single-step optimization preferable. In contrast, aligning with the nearest detection box in CycleTrack is unreliable, because targets may remain completely undetected. This leads to poor $D_{sim}$ estimation, explaining CycleTrack's inferior performance compared to CycleTrack-EM. An alternative perspective is that, compared to paired cyclic tasks, self-cyclic tasks allow for more stable EM training by eliminating the need to wait for one EM process to converge before initiating the next. We provide a more detailed analysis of the applicable scenarios in Appendix C.

**3. Can we rely solely on the cycle-consistency loss?**

Using only the $D_{cyc}$ loss in single-step optimization is mathematically equivalent to removing the E-step in the EM algorithm by eliminating the stop-gradient operation, which introduces a key limitation: the generated $\hat{y}$ often fails to adhere to the target domain or distribution due to the lack of explicit constraints. In our implementation, we disable the E-step by removing the forward-process freezing operation in CycleTrack-EM. As evidenced by Fig. 5, this ablated model converges to clearly trivial solutions compared to the complete EM training procedure. This behavior parallels the significant performance deterioration observed in CycleGAN when removing its adversarial loss while retaining only the cycle-consistency loss. This validates the necessity of both the $D_{sim}$ loss for single-step methods and the E-step for EM algorithms.

## 5 CONCLUSION

This work introduces a novel probabilistic framework that unifies cycle-consistent learning through variational modeling. By formulating cyclic tasks within a general theoretical foundation, we establish principled connections between previously disparate approaches. The framework naturally gives rise to two complementary optimization strategies – a VAE-style single-step method for efficient training, and an EM variant that operates without KL divergence estimation. Together, these contributions provide both theoretical coherence and practical flexibility for cycle-consistent learning across tasks. In image translation, our framework theoretically explains CycleGAN's success as a variational approximation of cycle-consistent learning, while enabling GAN-free alternatives via EM optimization. In object tracking, self-cyclic constraints enable dynamic modeling of target appearance variations, allowing both single-step and EM-based trackers to achieve robust unsupervised performance. We hope this work will inspire future research directions in cyclic learning.

### ACKNOWLEDGMENTS

This work was supported by the Guangdong S&T Program under Grant No. 2024B0101040005.

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

# A    EXTENDED INFERENCES

## A.1    FROM MAXIMIZING LOG-LIKELIHOOD TO MAXIMIZING ELBO

Here, we derive Eq. 3 in detail:

$$
\begin{aligned}
\log p_\theta(\mathbf{x}) &= \int q_\phi(\mathbf{y}|\mathbf{x}) \log p_\theta(\mathbf{x}) d\mathbf{y} \\
&= \int q_\phi(\mathbf{y}|\mathbf{x}) \log \frac{p_\theta(\mathbf{x}, \mathbf{y})}{p_\theta(\mathbf{y}|\mathbf{x})} d\mathbf{y} \\
&= \int q_\phi(\mathbf{y}|\mathbf{x}) \log \left( \frac{p_\theta(\mathbf{x}, \mathbf{y})}{q_\phi(\mathbf{y}|\mathbf{x})} \cdot \frac{q_\phi(\mathbf{y}|\mathbf{x})}{p_\theta(\mathbf{y}|\mathbf{x})} \right) d\mathbf{y} \\
&= \int q_\phi(\mathbf{y}|\mathbf{x}) \log \frac{p_\theta(\mathbf{x}, \mathbf{y})}{q_\phi(\mathbf{y}|\mathbf{x})} d\mathbf{y} + \int q_\phi(\mathbf{y}|\mathbf{x}) \log \frac{q_\phi(\mathbf{y}|\mathbf{x})}{p_\theta(\mathbf{y}|\mathbf{x})} d\mathbf{y} \\
&= \mathbb{E}_{q_\phi(\mathbf{y}|\mathbf{x})} \left[ \log \frac{p_\theta(\mathbf{x}, \mathbf{y})}{q_\phi(\mathbf{y}|\mathbf{x})} \right] + D_{KL}(q_\phi(\mathbf{y}|\mathbf{x}) || p_\theta(\mathbf{y}|\mathbf{x})),
\end{aligned}
\tag{18}
$$

with

$$
\begin{aligned}
\ell_{\theta,\phi}(\mathbf{x}) &= \mathbb{E}_{q_\phi(\mathbf{y}|\mathbf{x})} \left[ \log \frac{p_\theta(\mathbf{x}, \mathbf{y})}{q_\phi(\mathbf{y}|\mathbf{x})} \right] \\
&= \int q_\phi(\mathbf{y}|\mathbf{x}) \log \frac{p_\theta(\mathbf{x}|\mathbf{y}) p(\mathbf{y})}{q_\phi(\mathbf{y}|\mathbf{x})} d\mathbf{y} \\
&= \int q_\phi(\mathbf{y}|\mathbf{x}) \log p_\theta(\mathbf{x}|\mathbf{y}) d\mathbf{y} + \int q_\phi(\mathbf{y}|\mathbf{x}) \log \frac{p(\mathbf{y})}{q_\phi(\mathbf{y}|\mathbf{x})} d\mathbf{y} \\
&= \mathbb{E}_{q_\phi(\mathbf{y}|\mathbf{x})} \left[ \log p_\theta(\mathbf{x}|\mathbf{y}) \right] - D_{KL}(q_\phi(\mathbf{y}|\mathbf{x}) || p(\mathbf{y})).
\end{aligned}
\tag{19}
$$

The above derivation yields a decomposition of the log-likelihood $\log p_\theta(\mathbf{x})$ into a lower bound. Starting from the log-marginal likelihood, Eq. 18 introduces a variational distribution $q_\phi(\mathbf{y}|\mathbf{x})$ and decomposes $\log p_\theta(\mathbf{x})$ into the sum of two expectation terms: the first term is the evidence lower bound $\ell_{\theta,\phi}(\mathbf{x})$, and the second term is the KL divergence between the variational distribution $q_\phi(\mathbf{y}|\mathbf{x})$ and the true posterior $p_\theta(\mathbf{y}|\mathbf{x})$. Eq. 19 further expands the ELBO term $\ell_{\theta,\phi}(\mathbf{x})$ into its standard form: the difference between a reconstruction expectation term and a KL divergence term.

## A.2    ANOTHER CONSTRAINT: FEATURE CONSISTENCY

The feature consistency assumes that corresponding data points from the two domains can overlap when mapped to the same space, or that paired data samples share domain-invariant features. In essence, this serves as a way to constrain $\mathbf{y}$ by aligning $\mathbf{x}$ and $\mathbf{y} = f(\mathbf{x})$ in a particular feature space, thereby replacing or enhancing the role of $D_{KL}(q_\phi(\mathbf{y}|\mathbf{x})||p(\mathbf{y}))$ in single-step optimization. The difference between feature consistency and cycle consistency in constraining the range of $\mathbf{y}$ lies in their metrics: the former is represented by $D_{sim}(\mathbf{y}, \mathbf{x})$ while the latter is approximated by $D_{sim}(\mathbf{y}, \mathcal{Y})$. Since additional reasonable constraints are introduced, the mapping quality may improve as mentioned in the main text, but $D_{sim}(\mathbf{y}, \mathbf{x})$ may also lack proper definition in practical applications as is the case with $D_{sim}(\mathbf{y}, \mathcal{Y})$.

For instance, Wang et al. (2019b) models the relationship between forward and backward pixels as a self-cycle-consistent task, performing cyclic training across multiple frames with a one-step training pattern. In Eq. 16, $D_{cyc}$ is expressed in this work as the requirement for pixel alignment within the cycle formed by different frames, while $D_{sim}$ enforces feature consistency for the object across frames, serving as a replacement for the KL divergence. Dwibedi et al. (2019) proposes a self-supervised representation learning method which aims to learn frame-level embeddings by temporally aligning video sequences. This task is formulated as a *self-consistent task*. The method enforces $D_{cyc}$ by constructing bidirectional mappings between video frames. Meanwhile, $D_{KL}$ is implicitly satisfied as the same encoder maps inputs into the embedding space. Additionally, the mapping process in the embedding space inherently enforces a feature consistency constraint. Since feature consistency falls outside the scope of this work, we provide only a brief introduction here.

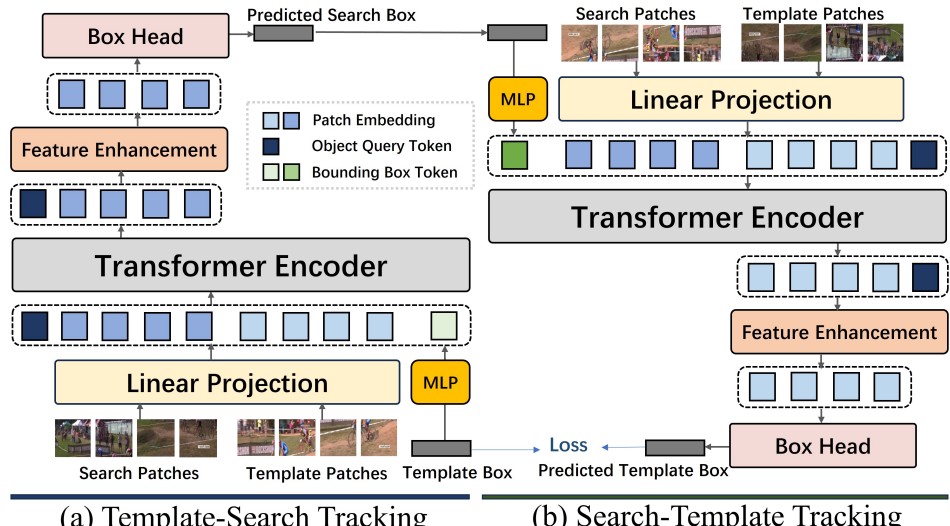

Figure 6: The structure of CycleTrack.

Table 7: The performance of different optimization methods in image translation tasks.

| Loss | Per-pixel accuracy | Per-class accuracy | Class IOU |
|---|---|---|---|
| $Labels \rightarrow Photo$ | | | |
| $D^{\mathcal{X}}_{cyc} + D^{\mathcal{X}}_{sim}$ | 0.41 | 0.07 | 0.03 |
| $D^{\mathcal{Y}}_{cyc} + D^{\mathcal{Y}}_{sim}$ | 0.39 | 0.05 | 0.02 |
| $D^{\mathcal{X}}_{cyc} + D^{\mathcal{X}}_{sim} \rightleftarrows D^{\mathcal{Y}}_{cyc} + D^{\mathcal{Y}}_{sim}$ | 0.51 | 0.13 | 0.10 |
| CycleGN | 0.52 | 0.14 | 0.10 |
| CycleGAN(Zhu et al., 2017) | 0.52 | 0.17 | 0.11 |
| $Photo \rightarrow Labels$ | | | |
| $D^{\mathcal{X}}_{cyc} + D^{\mathcal{X}}_{sim}$ | 0.10 | 0.06 | 0.02 |
| $D^{\mathcal{Y}}_{cyc} + D^{\mathcal{Y}}_{sim}$ | 0.32 | 0.09 | 0.05 |
| $D^{\mathcal{X}}_{cyc} + D^{\mathcal{X}}_{sim} \rightleftarrows D^{\mathcal{Y}}_{cyc} + D^{\mathcal{Y}}_{sim}$ | 0.52 | 0.16 | 0.11 |
| CycleGN | 0.51 | 0.16 | 0.10 |
| CycleGAN(Zhu et al., 2017) | 0.58 | 0.22 | 0.16 |

# B EXTENDED EXPERIMENTS

## B.1 SEVERAL VARIANTS OF SINGLE-STEP LOSS

This paper focuses on the joint optimization of $\max(\log p_\theta(\mathbf{x}) + \log p_\phi(\mathbf{y}))$. We show that optimizing only a single direction, specifically maximizing $p(\mathbf{x})$ alone or $p(\mathbf{y})$ alone, influences performance in both forward and backward tasks. Tab. 7 summarizes the results, where $D^{\mathcal{X}}_{cyc} + D^{\mathcal{X}}_{sim}$ and $D^{\mathcal{Y}}_{cyc} + D^{\mathcal{Y}}_{sim}$ correspond to maximizing $\log p(\mathbf{x})$ and $\log p(\mathbf{y})$ respectively. Another variant involves rotated optimization of $\log p(\mathbf{x})$ and $\log p(\mathbf{y})$, denoted in the table as $D^{\mathcal{X}}_{cyc} + D^{\mathcal{X}}_{\text{sim}} \rightleftarrows D^{\mathcal{Y}}_{cyc} + D^{\mathcal{Y}}_{sim}$. Unlike Algo. 1, this approach switches objectives every mini-batch without awaiting convergence between two EM cycles. While this configuration demonstrates inferior performance to CycleGAN on both tasks, it remains viable. We argue that under constrained training conditions, splitting Eq. 12 into dual cyclic objectives with rotated training provides a feasible alternative.

## B.2 FULLY-SUPERVISED AND SEMI-SUPERVISED EXPERIMENTS OF CYCLETRACK

CycleTrack was evaluated under identical settings to STARK for supervised training. In the semi-supervised setup, only the annotation of the target in the first frame of each sequence was available.

Table 8: Comparison between fully-supervised and semi-supervised CycleTrack with state-of-the-art fully-supervised trackers on LaSOT and TrackingNet.

| Method | LaSOT | | TrackingNet | |
|---|---|---|---|---|
| | AUC | Precision | AUC | Precision |
| DiMP(Bhat et al., 2019) | 56.9 | 56.7 | 74.0 | 68.7 |
| STARK(Yan et al., 2021) | 67.1 | - | 82.0 | - |
| Mixformer-22k(Cui et al., 2022) | 69.2 | 74.7 | 83.1 | 81.6 |
| OSTrack(Ye et al., 2022) | 69.1 | 75.2 | 83.1 | 82.0 |
| GRM(Gao et al., 2023) | 69.9 | **75.8** | **84.0** | **83.3** |
| CycleTrack (fully) | **70.1** | 75.2 | 82.9 | 81.0 |
| CycleTrack-EM (semi) | 63.6 | 66.7 | 80.6 | 78.3 |

Since the similarity loss $D_{sim}(\hat{\mathbf{y}}, \mathcal{Y})$ cannot be defined in search frames under this configuration, the single-step optimization method becomes inapplicable. Therefore, we exclusively employed the EM approach for semi-supervised training. As shown in Tab. 8, CycleTrack achieves comparable performance to leading two-frame trackers, demonstrating that our assembled network possesses the fundamental capabilities expected of a competent tracker. Notably, the semi-supervised CycleTrack-EM delivers remarkably strong performance, showing minimal degradation compared to its fully-supervised counterpart and even surpassing the fully-supervised DiMP. These results validate the effectiveness of our proposed training methodology.

## C  EXTENDED DISCUSSIONS

Given that the single-step approach is more conceptually straightforward and has achieved remarkable success in CycleGAN-related applications, while the EM-based method is less prevalent, we provide the following recommendations for its applicable scenarios:

**i. Scenarios without $D_{KL}$ estimation.**

GANs serve as an effective estimator of $D_{KL}$, but they are not universally suitable for all tasks. In practice, apart from GANs, few methods can reliably approximate KL divergence. While image-based domain adaptation largely adopts CycleGAN and uses GANs as the estimator, other modalities attempting to use single-step losses have generally achieved suboptimal results. This has compelled some studies to resort to alternatives such as feature consistency(discussed in Appendix A.2). Adopting EM-based methods may offer a new perspective for addressing problems in non-GAN scenarios.

**ii. Scenarios where convergence to local optima can be avoided.**

**a).** Semi-supervised settings with limited annotated data: Even a small amount of annotated data can prevent domains $\mathcal{X}$ and $\mathcal{Y}$ from forming independent cycles. In such configurations, domains $\mathcal{X}$ and $\mathcal{Y}$ are necessarily connected. Even if the network tends to generate simpler outputs, the locally optimal domains $\mathcal{B}$ and $\mathcal{A}$ will remain closer to the true domains $\mathcal{X}$ and $\mathcal{Y}$.

**b).** When $f_\phi$ and $g_\theta$ are already capable of generating outputs in the target domain: In this case, although the network lacks direct mapping ability between domains, it has acquired the capacity to generate results in domain $\mathcal{Y}$ through other upstream tasks (even if not starting from domain $\mathcal{X}$). EM training then acts as fine-tuning on a pre-trained network, significantly reducing the possibility of the network freely diverging to other domains.

**c).** When no alternative domains are available to the generative function—i.e., in strictly bijective scenarios: While training CycleTrack, we observed that for any output quadruple beyond object bounding boxes, there are hardly any other meaningful or structured combinations that can satisfy cycle consistency. This makes producing object bounding descriptions the most straightforward choice for the network.

**iii. Scenarios that call for a quick and simple trial.**

In practice, EM training involves only the reconstruction loss $D_{cyc}$, which is generally the simplest loss to design across tasks and modalities. In contrast, designing and optimizing $D_{sim}$ is consid-

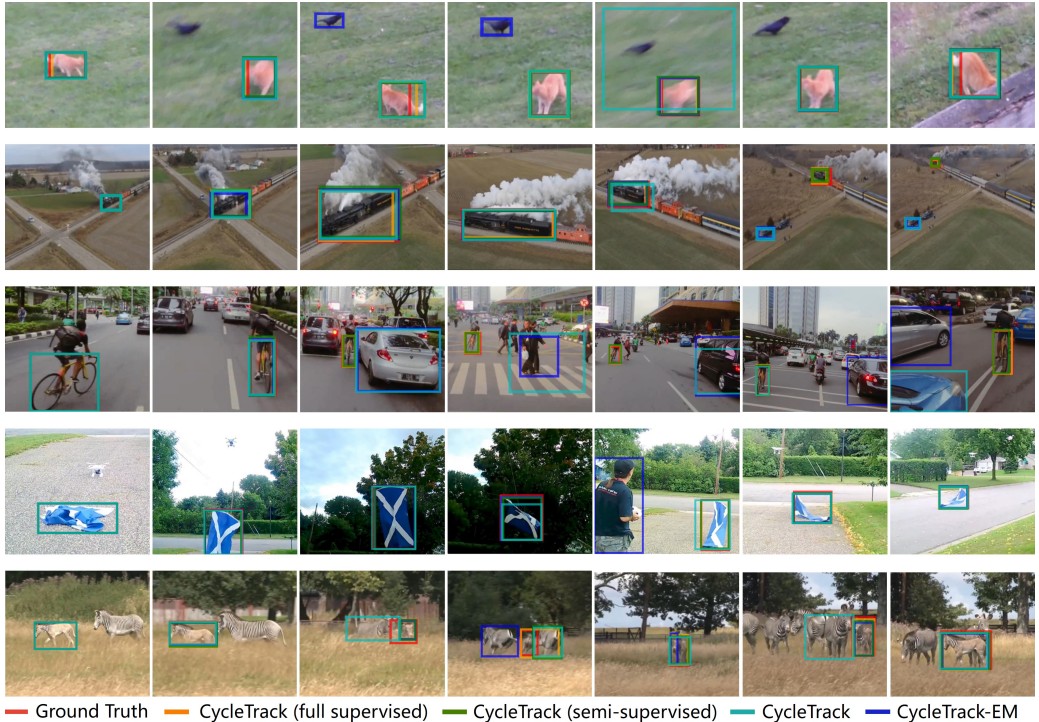

—— Ground Truth  —— CycleTrack (full supervised)  —— CycleTrack (semi-supervised)  —— CycleTrack  —— CycleTrack-EM

(a) Visualization of tracking results by different CycleTrack variants on LaSOT.

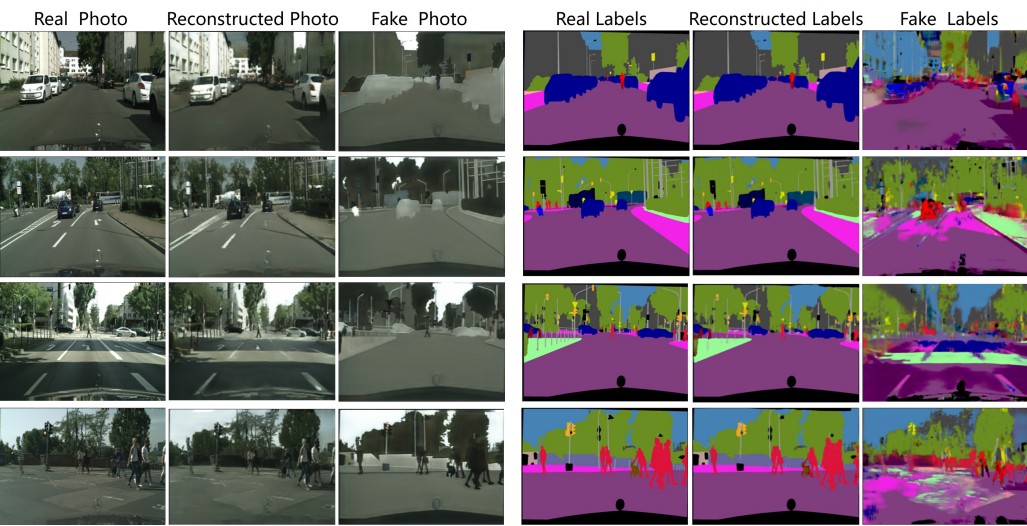

(b) Visualization of photo↔labels translation results by CycleGN on Cityscape.

erably more challenging. Another key consideration lies in the structural constraints imposed by single-step loss training: for $f_\phi$ and $g_\theta$ to be trained end-to-end, they must form a fully differentiable pipeline. Satisfying this requirement often places strong constraints on the architecture of both networks—sometimes even necessitating customized designs, as encountered in our work on CycleTrack. In contrast, the EM approach imposes no structural constraints on $f_\phi$ and $g_\theta$, since the two networks are trained separately. This enables the use of non-differentiable operations during training.

