# OpenReview forum: "Variational Inference for Cyclic Learning"
_ICLR.cc/2026/Conference — ICLR 2026 Poster_

### Official Review · Reviewer_w19x · 2025-10-26

**Soundness:** 3
**Presentation:** 2
**Contribution:** 3
**Rating:** 8
**Confidence:** 4

**Summary:**

This paper has a good intuition to use the latent variables bridging the two domains. They summarize the success of CycleGan by modeling the two terms of reconstruction and distribution alignment and propose several multi-variants like “CycleGN” (GAN-free)  and CycleTrack / CycleTrack-EM and achieve SOTA on unsupervised visual tracking tasks.

**Strengths:**

Sterngeths:


The paper establishes a framework to broad the cyclin-style loss functions in computer vision tasks. The paper has a clear concept by understanding the cycle term as reconstruction and the GAM term as the KL surrogate for making single-step optimization. Their proposed methods help address the failure conditions for the CycleGAN and are useful in the visual tracking tasks.

**Weaknesses:**

Weakness:


1 Overall the math deductions are good, but there are many typos and rigor issues, e.g.: In eq.3, where does the p_{data} come from?; Extra parenthesis in Eq. (16); what are the abbreviations IoU for in Eq. (17)? Should it be argmax based on the results in the table 3-4?


2 Figure 2 and Figure 3 are not clear enough to be understood.


3 CycleGN is close but generally behind CycleGAN in the more challenging direction (photo→labels), suggesting that the Dsim/EM recipe doesn’t yet match a well-trained discriminator as a KL surrogate.

**Questions:**

Questions:


1  What are the abbreviations IoU for in Eq. (17)? Should it be argmax based on the results in the table 3-4?


2 Could you explain in more detail when your method will be successful? Will your methods be useful to the questions like domain adaptations in different modalities?

---

> ### Author Response · Authors · 2025-11-19
> **Response to Reviewer w19x**
>
> Thank you for your comments and suggestions for revising the paper. We will address all the identified issues, and the updated version will be uploaded in the coming days.
>
> ---
>
> $\textbf{A1:}$
> Revisions to the paper: We acknowledge the lack of details in certain sections. In Eq. 17, IoU (Insertion over Union) measures how much bounding box $\textbf{x}$ belongs to an object in $\mathcal{X}$. The $\arg\max$ (Indeed, it should be $\arg\max$ instead of $\arg\min$; **we apologize for the error!**) finds the box $\tilde{\textbf{y}}$ that maximizes this IoU with the tracker's predicted box $\hat{\textbf{y}}$. Issues such as the definition of $p_{data}$ in Eq. 3 and extra parentheses in Eq. 16 will be carefully corrected in the new version. Additionally, citation formatting issues will be uniformly addressed. To improve clarity, we will remove the formulas from Figures 2 and 3 and provide a more accessible explanation within the figures themselves.
>
> ---
>
> $\textbf{A2:}$ Analysis of Applicable Scenarios
>
> While our direct comparison of single-step and EM methods is limited to two tasks, the broader relevance of cyclic learning is evident across industry and research. Here, we strive to use cross-modal tasks as illustrative examples:
>
> $\textbf{i. Scenarios without $D_{KL}$ estimation}$
>
> GANs serve as an effective estimator of $D_{KL}$, but they are not universally suitable for all tasks. In practice, apart from GANs, few methods can reliably approximate KL divergence. While image-based domain adaptation largely adopts CycleGAN and uses GANs as the estimator, other modalities attempting to use single-step losses have generally achieved suboptimal results. This has compelled some studies to resort to alternatives such as feature consistency, as discussed in our Appendix A.2. Adopting EM-based methods may offer a new perspective for addressing problems in non-GAN scenarios.
>
> $\textbf{ii. Scenarios where convergence to local optima can be avoided}$
>
> As analyzed in our **response to uEwd, A2** and **response to dfXf, A1**, EM methods may converge to  a suboptimal solution comprising two specific mapping pairs. However, this can be mitigated in practical applications:
>
> **a).** Semi-supervised settings with limited annotated data:
> Even a small amount of annotated data can prevent domains $\mathcal{X}$ and $\mathcal{Y}$ from forming independent cycles. In such configurations, domains $\mathcal{X}$ and $\mathcal{Y}$ are necessarily connected. Even if the network tends to generate simpler outputs, the “lazy” domains B and A will remain closer to the true domains $\mathcal{X}$ and $\mathcal{Y}$. For example, CyCO—a semi-supervised method based on EM—performs cyclic learning for visual grounding and regional image captioning. With only 20\% supervised data, it achieves performance close to that of fully supervised approaches.
>
> **b).** When $f_\phi$ and $g_\theta$ are already capable of generating outputs in the target domain:
> In this case, although the network lacks direct mapping ability between domains, it has acquired the capacity to generate results in domain $\mathcal{Y}$ through other upstream tasks (even if not starting from domain $\mathcal{X}$). EM training then acts as fine-tuning on a pre-trained network, significantly reducing the possibility of the network freely diverging to other domains. For instance, SC-Tune applies the EM method to Large Vision-Language Models for reference expression comprehension and generation, improving the model's adaptability to specific datasets.
>
> **c).** When no alternative domains are available to the generative function—i.e., in strictly bijective scenarios:
> While training CycleTrack, we observed that for any output quadruple beyond object bounding boxes, there are hardly any other meaningful or structured combinations that can satisfy cycle consistency. This makes producing object bounding descriptions the most straightforward choice for the network.
>
> $\textbf{iii. EM as a quick and easy choice}$
>
> In practice, EM training involves only the reconstruction loss $D_{cyc}$, which is generally the simplest loss to design across tasks and modalities. In contrast, designing and optimizing $D_{sim}$ is considerably more challenging.
> Another key consideration lies in the structural constraints imposed by single-step loss training: for $f_\phi$ and $g_\theta$ to be trained end-to-end, they must form a **fully differentiable** pipeline. Satisfying this requirement often places strong constraints on the architecture of both networks—sometimes even necessitating customized designs, as encountered in our work on CycleTrack.
> In contrast, the EM approach imposes no structural constraints on $f_\phi$ and $g_\theta$, since the two networks are trained separately. This enables the use of non-differentiable operations such as token indexing or Top-K selection during training.
> In summary, as a quick and easy-to-implement option, EM-based cyclic learning can be prioritized as a tool for domain adaptation.

---

> > ### Comment · Reviewer_w19x · 2025-11-23
> >
> > Thanks for your responses. I am happy to give this paper accept.

---

### Official Review · Reviewer_dfXf · 2025-11-01

**Soundness:** 4
**Presentation:** 4
**Contribution:** 3
**Rating:** 8
**Confidence:** 2

**Summary:**

This paper proposes a unified probabilistic framework to generalize cyclic learning, moving beyond the ad-hoc, task-specific implementations that currently dominate the field. The authors identify that while cycle-consistency is a powerful tool for weakly-supervised learning, it lacks a common theoretical foundation. To address this, they reformulate the cycle-consistency objective as a variational inference (VI) problem. The core of their approach is to model the cross-domain mappings (e.g., $A \rightarrow B$ and $B \rightarrow A$) as conditional probability functions and treat the intermediate generated data as latent variables. This allows them to re-cast the training objective as the optimization of an Evidence Lower Bound (ELBO) on the data log-likelihood.

From this single theoretical framework, the authors derive two distinct and general optimization strategies. The first is a "VAE-style" single-step loss that optimizes the full objective at once, which the authors show provides a new theoretical justification for the success of architectures like CycleGAN. The second is an "EM-style" alternating optimization algorithm that iteratively updates the forward and backward mappings, avoiding the need for an explicit KL divergence approximation (like a GAN's discriminator). The framework's effectiveness is demonstrated on two very different tasks: unpaired image translation, where their proposed GAN-free "CycleGN" is competitive with CycleGAN, and unsupervised visual object tracking, where their "CycleTrack" and "CycleTrack-EM" models establish a new state-of-the-art on multiple benchmarks.

**Strengths:**

- Novel Theoretical Contribution: Its primary strength is the novel and elegant reformulation of cycle-consistency as a variational inference (VI) problem. This connects a widely-used heuristic to fundamental probabilistic principles (like ELBO maximization, VAEs, and the EM algorithm) for the first time.

- Generalization: The framework is highly general, providing a unified theory for both paired cyclic tasks (like image translation) and self-cyclic tasks (like video tracking), which were previously treated with separate, task-specific methods.

**Weaknesses:**

- Analysis of EM-Style Failure Modes: The paper honestly presents failure cases (Fig. 4a) where the EM-based CycleGN achieves good cycle-reconstruction but produces poor-quality intermediate "fake" images. This suggests the model has learned an "incorrect mapping" that satisfies $g(f(x)) \approx x$ but where $f(x)$ is not a faithful member of the target domain $Y$. This is a crucial finding and a known risk of EM-style approaches converging to a local optimum. The paper would be improved by a deeper analysis of why this happens in the VI framework. Is it an inherent instability of the alternating optimization? Or does it confirm that the EM approach, by "lacking explicit constraints on latent variables" (Section 2.1), is more vulnerable to this specific failure mode than the single-step method, which explicitly constrains the intermediate variable with $D_{sim}$?

- Limited Scope of Image Translation Experiments: The validation of CycleGN is performed only on the Cityscapes (labels $\leftrightarrow$ photo) dataset. This is a highly structured translation task. The original CycleGAN paper demonstrated its robustness on much more "unstructured" and challenging tasks, such as horse $\rightarrow$ zebra or style transfer (Monet $\rightarrow$ photo). To compellingly claim CycleGN is a general, competitive alternative to CycleGAN, it should be tested on these more diverse and difficult translation tasks. It is possible the EM-style approach works well for structured tasks but struggles with more unconstrained mappings where a GAN's distributional matching is essential.

**Questions:**

Regarding the CycleGN failure case in Figure 4a (where reconstruction is good but intermediate images are poor), could you elaborate on the cause? Is this a local optimum that is inherent to the alternating EM optimization, or does this failure mode confirm that the EM approach is more vulnerable to "cheating" precisely because it lacks the explicit $D_{sim}$ (distributional) constraint on the latent variable that the single-step method has?

The claim that CycleGN is a general, competitive alternative to CycleGAN would be significantly strengthened by testing it on more unstructured translation tasks (e.g., horse $\leftrightarrow$ zebra, style transfer). Have you performed experiments on such tasks? How does the EM-style approach perform in these more unconstrained settings where the GAN-based $D_{sim}$ term is known to be critical?

---

> ### Author Response · Authors · 2025-11-19
> **Response to Reviewer dfXf**
>
> Thank you for your comments. Your questions regarding the failure cases are highly valuable for understanding EM-based iterative learning. They have enabled us to conduct a more insightful analysis of the limitations of the EM method, which will be incorporated into the main text soon.
>
> ---
>
> $\textbf{A1:}$ Failure Case: We attribute this issue to an inherent local optimum in EM-based training: during learning, $g_\theta$ adapts to two distinct modes—$g_\theta(a) = \textbf{x}$ and $g_\theta(\textbf{y}) = b$—thus learning both A→$\mathcal{X}$ and $\mathcal{Y}$→B mappings using the same $\theta$. Accordingly, $f_\phi$ learns $\mathcal{X}$→A and B→$\mathcal{Y}$, with domains A and B emerging from network "laziness". This leads to unintended bidirectional mappings A$\leftrightarrow\mathcal{X}$ and $\mathcal{Y}\leftrightarrow$B. For clarity, we explain the underlying rationale alongside the domain constraints in our **response to uEwd, A2**.
> Once this steady state is reached, the E-step assumption $p_{data}(\textbf{y}|\textbf{x}) = p_\phi(\textbf{y}|\textbf{x})$ becomes invalid, revealing inherent instability in alternating optimization.
>
> In CycleGN, this yields high-quality reconstructions but poor intermediate images—though some patterns persist (e.g., uniform car color in "fake photos"). We interpret this as $f_\phi$ converging to domain A, which is close to but distinct from $\mathcal{Y}$.
>
> That said, domains A and B are not inevitable. When explicit distributional constraints are imposed, $g_\theta$ and $f_\phi$ are compelled to remain within $\mathcal{X}$ and $\mathcal{Y}$. This is because with $D_{sim}$, the constraint on the output domain is explicit, whereas the assumption $p_{data}(\textbf{y}|\textbf{x}) = p_\phi(\textbf{y}|\textbf{x})$ imposes only an implicit constraint on the domain—a assumption that is potentially unreliable. This is ultimately why we argue that such local optima stem from the lack of explicit constraints on latent variables.
>
> ---
>
> $\textbf{A2:}$ Further Experiments: We conducted tasks similar to horse$\leftrightarrow$zebra, such as orange$\leftrightarrow$apple and sunmmer$\leftrightarrow$winter. To be honest, CycleGN performs significantly weaker than CycleGAN on these tasks. This is partly because hyperparameters should be carefully selected for different problems, whereas we applied the same settings used for the photo$\leftrightarrow$label task to the above tasks, such as when to terminate an inner loop. However, it is unnecessary to hide the fact that training such tasks is inherently more challenging.
>
> Regarding this phenomenon, the main questions we are concerned with are:
> Why is it more difficult to train one type of problem compared to another, even when both use EM alternating training?
> Why does the EM algorithm for unstructured image generation struggle more to converge to the target domain compared to structured image generation?
>
>
> We believe the primary reason is that the EM method relies more heavily on the assumption of Dirac distributions than GAN-like methods—that is, it requires the mapping between domain $\mathcal{X}$ and domain $\mathcal{Y}$ to be a **strict bijection**. As shown in our **response to uEwd, A2**, in addition to the derivation from the probabilistic model to the generative model, the EM method reintroduces the Dirac distribution assumption when using $\arg\max E_{p_{data}(\textbf{x}|\textbf{y})}[\log p_\phi(\textbf{y}|\textbf{x})]$ to ensure $\arg\min D_{KL}(p_{\phi}(\textbf{y}|\textbf{x})||p_{data}(\textbf{y}|\textbf{x}))$. This assumption strictly demands a one-to-one correspondence between $\textbf{x}$ and $\textbf{y}$. However, such a strict condition is often difficult to satisfy in unstructured image generation. Taking horse$\leftrightarrow$zebra as an example, horses and zebras each encompass a variety of possible appearances and are not inherently strict bijections. In practice, the mapping easily becomes many-to-many. Therefore, when faced with loose or imprecise correspondences, although both GAN-like and EM-based methods exhibit a certain degree of tolerance, the EM method demonstrates higher vulnerability due to its dependence on strict assumptions.
>
> Another interesting perspective is that, unlike label-generated tasks, these unstructured tasks lack strict quantitative metrics for evaluating quality. The absence of such metrics indicates that the target is more abstract and lacks a precise, unique definition. This observation actually points to a common reason why EM methods are more prone to failure in such scenarios: in these tasks, there is no single "correct" output for the generative model.
>
> Despite the aforementioned challenges, EM methods are not entirely outperformed by single-step approaches in practical applications. If you are interested in the scenarios where EM methods are applicable, you may refer to our **response to w19x, A2**.

---

### Official Review · Reviewer_uEwd · 2025-11-04

**Soundness:** 2
**Presentation:** 3
**Contribution:** 2
**Rating:** 4
**Confidence:** 4

**Summary:**

This paper presents a general framework for training cyclic learning models, built on single-step and alternating optimization strategies. The framework's broad utility is shown in two areas: for unpaired image translation, it not only provides a theoretical basis for CycleGAN but also produces CycleGN and CycleTrack, a competitive model that doesn't require GANs. Overall, this work lays the theoretical groundwork for cyclic learning and offers a universal approach for subsequent research.

**Strengths:**

The paper presents a method (CycleGN) that avoids the bottleneck of the previously proposed cycle methods, like unstable adversarial optimization (GAN-style). This enables competitive results without explicitly using discriminators. The paper is well-structured and easy to follow.

**Weaknesses:**

1) The performance of the *CycleGN EM-based* approach was found to be *inconsistent across tasks* - showing stable results in some settings but degraded performance in others compared to CycleGAN (single-step with GAN).  This indicates that method still lack a unified mechanism to fully remove the D_{KL} surrogate across different problem domains.


2) A issue remains where models can achieve nearly perfect while the *intermediate translation remains unrealistic. This is an intrinsic limitation of cyclic training.

**Questions:**

**Q1** : The connection between the proposed EM-style framework and the classical EM algorithm remains unclear.
    In the traditional EM formulation, the objective is to maximize the data likelihood.
    Could the authors clarify how optimizing the cycle-consistency loss ($D_{cyc}$) in their setting corresponds to maximizing the likelihood?
    Is it correct to interpret the cycle loss as a *surrogate reconstruction objective* that implicitly optimizes the likelihood function, similar to the role of the reconstruction term in variational inference?


 **Q2** :    Given the absence of explicit $D_{KL}$ control in the EM method, are there any theoretical or empirical evaluations confirming that the generated distributions $\mathcal{X}$ and $\mathcal{Y}$ approach the true priors, rather than merely achieving a cyclically consistent but unrealistic local optimum?


**Q3** :   Given the observed instability of results across different tasks, could the authors identify specific problem types or conditions under which the proposed EM-based framework is expected to outperform standard GAN-based models?
    What are the authors’ insights regarding the comparative advantages of their approach?
    In other words, what would constitute the \textit{ideal use case} where this framework would be preferable to a conventional GAN setup?

---

> ### Author Response · Authors · 2025-11-19
> **Response to Reviewer uEwd (1/2)**
>
> Thank you for your comment.  Given that $\textbf{Q2}$ raises a fundamental question, we will address it first ($\textbf{A2}$) before turning to the other points.
>
> ---
>
> $\textbf{A2:}$
> This is a key question regarding domain constraints and potential failures in our EM method. In the main text (Lines 234-238), we briefly addressed this from a network training perspective. Here we provide detailed elaboration and an additional theoretical explanation.
>
> $\textbf{i. Intuitive Explanation}$
>
> The cyclic learning process starts with a data point $\textbf{x}$,
>  generating $\hat{\textbf{y}} = f_\phi(\textbf{x})$ while $\phi$ remains frozen.
> This $\hat{\textbf{y}}$, which may initially fall outside domain$\mathcal{Y}$, serves as input to the trainable component $g_\theta$. Here is the key insight: although $g_\theta$ receives an incorrect input $\hat{\textbf{y}}$, it is trained to reconstruct the original $\textbf{x} \in \mathcal{X}$. This means that early in training, $g_\theta$ effectively learns to ignore its input and map anything back into domain $\mathcal{X}$. In contrast, the $E_\phi$-$M_\phi$ phase trains $f_\phi$ to enforce outputs within $\mathcal{Y}$. As training progresses,  $f_\phi$ produces higher-quality outputs, which in turn provide better inputs to $g_\theta$. Gradually, $g_\theta$ shifts from mapping arbitrarily to $\mathcal{X}$ toward learning the meaningful domain mapping from $\mathcal{Y}$ to $\mathcal{X}$, thereby achieving the cyclic learning objective.
>
> This explanation points to a potential failure mode: during training, $g_\theta$ might learn two mappings at the same time, $g_\theta(a) = \textbf{x}$ and $g_\theta(\textbf{y}) = b$, using the same parameters $\theta$. Meanwhile, $f_\phi$ would learn $\mathcal{X}$→A and B→$\mathcal{Y}$. As a result, the model ends up with bidirectional mappings between $\mathcal{X}\leftrightarrow$A and $\mathcal{Y}\leftrightarrow$B, which is not what we intended. This is why we noted that "the EM method does carry a risk of converging to local optima" (Line 241).
>
> $\textbf{ii. Theoretical Explanation}$
>
> **a).** Let's begin with the $\textbf{x}$→$\hat{\textbf{y}}$→$\textbf{x}$ cycle: In the $E_\theta$-$M_\theta$ step, $p_{data}(\textbf{y}|\textbf{x})$ is assumed known (i.e. $p_{data}(\textbf{y}|\textbf{x}) = p_{\phi*}(\textbf{y}|\textbf{x})$), setting the $D_{KL}$ term in Line 720 to zero. The objective simplifies to maximizing $\log p(\textbf{x})$, equivalent to maximizing its ELBO:
>
> $$
> \log p_\theta(\textbf{x}) \ge D_{KL}(p_{\phi*}(\textbf{y}|\textbf{x})||p_{data}(\textbf{y}|\textbf{x})) +\underbrace{E_{p_{\phi*}(\textbf{y}|\textbf{x})}[\log p_\theta(\textbf{x}|\textbf{y})]- D_{KL}(p_{\phi*}(\textbf{y}|\textbf{x})||p(\textbf{y}))}_{\text{ELBO}}.
> $$
>
> In this inner loop, both  $p_{\phi*}(\textbf{y}|\textbf{x})$ and $p(\textbf{y})$ are fixed, making the $D_{KL}(p_{\phi*}(\textbf{y}|\textbf{x})||p(\textbf{y}))$ term constant. Thus, the objective reduces to maximizing the expectation term only, i.e., optimizing $p_\theta(\textbf{x}|\textbf{y})$ to best reconstruct $\textbf{x}$ given the fixed $p(\textbf{y}|\textbf{x})$.
>
> **b).** During the $E_\phi$-$M_\phi$ phase, the $\textbf{y}$→$\hat{\textbf{x}}$→$\textbf{y}$ cyclic optimization is performed. Similarly, the objective at this stage is to maximize the expectation $E_{p_{\theta*}(\textbf{x}|\textbf{y})}[\log p_\phi(\textbf{y}|\textbf{x})]$.
>
> $\textbf{Lemma.}$ If $p(\textbf{x}|\textbf{y})$ and $p(\textbf{y}|\textbf{x})$ are Dirac distributions, then:
> $$
> \arg\max E_{p_{data}(\textbf{x}|\textbf{y})}[\log p_\phi(\textbf{y}|\textbf{x})] = \arg\min D_{KL}(p_{\phi}(\textbf{y}|\textbf{x})||p_{data}(\textbf{y}|\textbf{x})).
> $$
> *Proof.*
>  Simplifying the left side yields $E_{p_{data}(\textbf{x}|\textbf{y})}[\log p_\phi(\textbf{y}|\textbf{x})] = \log p_\phi(\textbf{y}|g(\textbf{y}))$. Thus, the left side is equivalent to performing maximum likelihood estimation on the dataset $\{g(\textbf{y}), \textbf{y}\}$.
> Since $p_{data}(\textbf{y}|\textbf{x})=\delta(\textbf{y}- f_\phi(\textbf{x}))$, minimizing the KL divergence forces $p_\phi(\textbf{y}|\textbf{x})$ to concentrate near $f_\phi(\textbf{x})$. Given the bijective condition $f_\phi(g(\textbf{y})) = \textbf{y}$, the objectives of both sides align: The left side aims to make $p_\phi(\textbf{y}|g(\textbf{y}))$ assign high probability to the true $\textbf{y}$, while the right side aims for it to assign high probability to $f_\phi(g(\textbf{y})) = \textbf{y}$.
> □
>
> It can be observed that $D_{KL}(p_{\phi}(\textbf{y}|\textbf{x})||p_{data}(\textbf{y}|\textbf{x}))$ is minimized in this phase.

---

> ### Author Response · Authors · 2025-11-19
> **Response to Reviewer uEwd (2/2)**
>
> **c).** Conclusion:
> The maximization of ELBO $\ell_\theta(\textbf{x})$ is achieved in the $E_\theta$-$M_\theta$ phase using $D_{\text{cyc}}^\mathcal{X}$, while the minimization of $\log p(\textbf{x}) - \ell_\theta(\textbf{x})$ is implemented in the $E_\phi$-$M_\phi$ phase. Conversely, the maximization of ELBO $\ell_\phi(\textbf{y})$ is achieved in the $E_\phi$-$M_\phi$ phase using $D_{\text{cyc}}^\mathcal{Y}$, while the minimization of $\log p(\textbf{y}) - \ell_\phi(\textbf{y})$ is implemented in the $E_\theta$-$M_\theta$ phase.
> From a macro perspective, the outer loop can be viewed as an alternating optimization algorithm for the ELBO and the KL divergence of the posterior distributions. It maximizes the ELBO in one EM phase while minimizing the $D_{KL}$ in the other, cycling through these steps to ultimately achieve the goal of maximizing the log-likelihood. Here, the $D_{KL}$ term constrains the generative functions to reside within the true domains $\mathcal{X}$ and $\mathcal{Y}$.
>
>
> The failure case in $\textbf{i}$ arises when the underlying assumptions are not satisfied. In the EM process, we assume that $p_{data}(\textbf{x}|\textbf{y})$ is ideal. In the example from $\textbf{i}$, if the two steady-state cycles of $X \leftrightarrow$ A and $Y \leftrightarrow$ B coexist simultaneously, it will cause the assumptions about $p_{data}$ in both EMs to perpetually fail. Therefore, it is necessary to carefully select hyperparameters and initial parameters, as well as the exit strategy for the inner loop.
>
> ---
>
> $\textbf{A1:}$ In Section 2 of the paper, we provided a detailed derivation from maximizing likelihood to maximizing $D_{cyc}$. Now we offer a step-by-step version to facilitate understanding.
>
> * **Step 1.** By introducing the variational variable $\textbf{y}$, maximizing $\log p(\textbf{x})$ can be transformed into maximizing the ELBO. (Eqs. 3, 4 and 5 in the paper.)
> * **Step 2.** Under a bijective condition (conforming to the Dirac distribution), maximizing the ELBO is equivalent to maximizing $D_{cyc} + D_{sim}$. (Eqs. 1 and 2 for the Dirac distribution; Eqs. 7 and 8 for $D_{cyc}$; Eqs. 9 and 10 for $D_{sim}$; Eqs. 11 and 12 as a summary.)
> * **Step 3.** In the alternating EM algorithm, $D_{sim}$ ($D_{KL}$) remains constant; therefore, only maximizing $D_{cyc}$ needs to be considered. (Lines 216-222, a comprehensive explanation is given in $\textbf{A2}$.)
>
> Connection to the Naive EM Algorithm:
>
> The naive EM algorithm performs coordinate ascent by alternately fixing parameters $\theta$ to find the optimal latent distribution $q(z)$ (E-step), and then updating $\theta$ to maximize the expected likelihood under $q(z)$ (M-step).
> Our method adopts this alternating structure but targets the learning of mappings $p(\textbf{x}|\textbf{y})$ and $p(\textbf{y}|\textbf{x})$. For $p_\theta(\textbf{x}|\textbf{y})$, the E-step fixes $\theta$ (via $E_\phi$) and finds $p_\phi(\textbf{y}|\textbf{x})$ (via $M_\phi$) to tighten the ELBO; the M-step then maximizes $p_\theta(\textbf{x}|\textbf{y})$'s expected likelihood(via $M_\theta$) under $q_\phi(\textbf{y}|\textbf{x})$ (via $E_\theta$). The same logic applies symmetrically to $p_\phi(\textbf{y}|\textbf{x})$.
> It can be seen that our method is also entirely a coordinate ascent approach, alternately optimizing the probabilistic model parameters and the distribution function, with the key distinction that the distribution functions themselves are parameterized.
>
> ---
>
> $\textbf{A3:}$ We share the same concern regarding the practical choice of the EM algorithm. As detailed in our response to **Reviewer w19x**, its suitability depends on the application.
> Admittedly, in GAN-specific scenarios, EM methods are unlikely to outperform GANs, as the latter inherently provide a robust estimator for $D_{KL}$ through joint optimization, which reduces additional assumptions and local optima risks. However, as our analysis indicates, GANs are not suitable for all modalities, and single-loss optimization via **differentiable connections** between $f_\phi$ and $g_\theta$ is not a free lunch either. In such cases, the alternating EM method offers a simple and efficient implementation for cyclic learning.
>
> ---
>
> $\textbf{Response to weakness:}$: We acknowledge the noted intrinsic limitations of both the alternating EM method and cyclic learning. While our cycle consistency-based solution is imperfect, we believe an inherent gap between unsupervised and supervised performance is unavoidable without paired labels.

---

### Meta-Review · Area_Chair_sQcM · 2026-01-17

**Summary:**

This paper proposes a unifying theoretical framework for cyclic learning by casting cycle-consistency objectives as ELBO optimization under a variational inference view, and derives two general training strategies: single-step and alternating/EM-style optimization. It argues this perspective both (i) provides a principled justification for CycleGAN-style objectives and (ii) yields a GAN-free alternative (CycleGN) as well as cyclic-learning instantiations for unsupervised tracking (CycleTrack / CycleTrack-EM). Overall, the contribution is primarily conceptual/theoretical (a general lens on cyclic learning) with supporting empirical demonstrations across multiple tasks.

The overall ratings are 8,8,4. With the 4 being potentially improved, the AC recommends acceptance. While some empirical generality concerns remain (notably evaluations on harder unstructured translation tasks and robustness across domains), the overall contribution and reviewer support (two accepts vs. one borderline reject driven largely by clarity concerns that were addressed) justify acceptance.

**Reviewer Concerns:**

Concerns substantially addressed:
- Clarifying the EM interpretation / likelihood connection and the role of cycle loss: Reviewer uEwd explicitly questioned the connection to classical EM and whether optimizing cycle loss corresponds to likelihood/VI objectives. The authors’ rebuttal provides a much more detailed explanation (including an intuitive narrative of the alternating phases and a formal derivation sketch), which meaningfully improves interpretability of the “EM-style” framing.
- Explaining failure modes / domain-constraint issues (local optima, “cheating”): Both uEwd and dfXf asked about cases where reconstruction is good but intermediate translations are unrealistic and whether the EM-style method is more vulnerable without explicit distributional constraints. The rebuttal directly engages this, describing how alternating phases can fail (e.g., degenerate mappings / competing cycles) and acknowledges the risk of local optima, which is a concrete improvement over the initial presentation.

Concerns still outstanding / only partially addressed:
- Empirical breadth on unstructured image translation tasks: Reviewer dfXf requested evaluation on harder, more diverse unpaired translation tasks (e.g., horse↔zebra, style transfer) to support the “general competitive alternative to CycleGAN” claim. This point remains largely unresolved in the discussion record: the core request is empirical, and the rebuttal as captured here mainly expands analysis rather than adding those missing experiments.
- Inconsistent performance across tasks / limits of removing the KL surrogate: Both uEwd and dfXf emphasize that the EM-based approach can be inconsistent across settings and may trail CycleGAN in challenging directions. The rebuttal acknowledges intrinsic limitations, but the underlying concern (robustness + when/why it wins) remains only partially settled without broader empirical validation.

**Reviewer Scores:**

- Reviewer uEwd (initial: 4, marginally below threshold): uEwd’s main blockers were conceptual clarity (EM connection, likelihood interpretation, failure cases / domain constraints) and the “ideal use case” framing. Given the rebuttal’s substantial added explanation and explicit acknowledgment of failure modes, I expect uEwd would move up modestly, likely to 6 (borderline accept)—still with reservations due to remaining empirical breadth/robustness concerns.

- Reviewer dfXf (initial: 8, accept): likely stay 8

- Reviewer w19x (initial: 8, accept): likely stay 8

---

### Decision · Program_Chairs · 2026-01-26

Accept (Poster)